# Emergence of enhancers at late DNA replicating regions

Paola Cornejo-Páramo[1,2], Veronika Petrova[1,2], Xuan Zhang [1], Robert S. Young [3,4] & Emily S. Wong [1,2] ✉

Enhancers are fast-evolving genomic sequences that control spatiotemporal gene expression patterns. By examining enhancer turnover across mammalian species and in multiple tissue types, we uncover a relationship between the emergence of enhancers and genome organization as a function of germline DNA replication time. While enhancers are most abundant in euchromatic regions, enhancers emerge almost twice as often in late compared to early germline replicating regions, independent of transposable elements. Using a deep learning sequence model, we demonstrate that new enhancers are enriched for mutations that alter transcription factor (TF) binding. Recently evolved enhancers appear to be mostly neutrally evolving and enriched in eQTLs. They also show more tissue specificity than conserved enhancers, and the TFs that bind to these elements, as inferred by binding sequences, also show increased tissue-specific gene expression. We find a similar relationship with DNA replication time in cancer, suggesting that these observations may be time-invariant principles of genome evolution. Our work underscores that genome organization has a profound impact in shaping mammalian gene regulation.

Enhancers are *cis*-regulatory elements essential as modulators of spatiotemporal gene expression by acting as integrators of trans-acting signals by recruiting transcription factors (TFs) and other effector molecules. Enhancers are typically rapidly evolving and are frequently species-specific[1–4]. For example, most human enhancers are not found in the mouse[3].

The factors responsible for enhancer turnover are not well understood. The prevailing model of enhancer evolution is the mobilization of transposable elements (TE) and their insertions to new genomic locations[5]. As TEs are commonly found overlapping enhancers, they have been hypothesized to play a major role in the dynamic landscape of enhancer turnover in mammals by distributing cis-regulatory elements across the genome. However, they do not account for most recently evolved mammalian enhancers, many of which appear to have originated from ancestral sequences without prior biochemical activity in the same tissue[6–10].

Notably, local point mutations in non-regulatory sequences can give rise to enhancer activity, suggesting an alternative mechanism for generating tissue-specific enhancers[11–16]. One of the most significant predictors of local mutational density is DNA replication time[17–19]. De novo mutations are elevated at late replicating genome regions corresponding to the latter stages of S-phase - a phenomenon that is potentially due to a reduced ability of DNA repair mechanisms to function effectively during late replication time[20]. In hominids and rodents, mutation rates are 20–30% higher at late compared to early replication domains[21], and this trend of accelerated mutational rate extends throughout eukaryotic evolution[22]. The DNA replication timing program, defined by the temporal order of DNA replication during the S-phase, is also closely linked to the spatial organization of chromatin in the nucleus and transcriptional activity[23]. Late replicating domains are associated with facultative heterochromatin and tissue-specific gene expression[24,25].

[1]Victor Chang Cardiac Research Institute, Darlinghurst, NSW, Australia. [2]School of Biotechnology and Biomolecular Sciences, Sydney, NSW, Australia. [3]Usher Institute, University of Edinburgh, Teviot Place, Edinburgh EH8 9AG, United Kingdom. [4]Zhejiang University - University of Edinburgh Institute, Zhejiang University, 718 East Haizhou Road, 314400 Haining, PR China. ✉e-mail: e.wong@victorchang.edu.au

Thus, we hypothesized that DNA replication timing plays a role in the emergence and diversification of enhancers through de novo mutations[2,26,27]. In the context of enhancer turnover in mammals, we investigate the role of nucleotide substitutions linked to DNA replication timing. Using detailed maps of candidate cis-regulatory elements across species, we take a multi-scale approach to explore the relations between enhancer turnover and the genome. We examine the contribution of TEs and the de novo creation of TF binding sites to enhancer turnover. By comparing enhancers across DNA replication domains and their tissue-specific activity across vastly different time scales, we aim to illuminate the evolutionary trajectories of enhancers and their implications for gene regulation.

## Results

### Germline replication time is associated with the rate of enhancer turnover across the genome

Genetic changes occurring in the germline provide genetic variation that is the substrate for species evolution. We examined multi-tissue enhancer turnover in mice comparing across germline DNA replication time. Evolutionarily conserved and recently evolved, i.e., lineage-specific mouse enhancers were annotated using histone mark ChIP-seq data based on multi-species comparisons (cat, dog, horse, macaque, marmoset, opossum, pig, rabbit, and rat)[9]. Following convention, candidate enhancers are defined as sequences enriched for H3K27ac but absent in H3K4me3 (termed "active") or enriched for H3K4me1 (termed "poised")[9,28] (Fig. 1A, B). To ensure robustness, all enhancers were identified using consensus regions defined by overlapping multiple biological replicates by a minimum of 50% of their length[9].

Next, enhancers were annotated as evolutionarily conserved if they possess enhancer-associated histone marks in at least two other species ($n = 94,107$). Recently evolved enhancers were defined as cis-regulatory elements identified only in mice ($n = 80,904$), where approximately half of these regions aligned to non-regulatory regions in other species (~49%; liftOver -minMatch = 0.6). This supports similar findings in human enhancers[3]. Both conserved and species-specific enhancers showed a similar propensity to overlap ATAC-seq peaks, indicating comparable levels of chromatin accessibility (Methods, Supplementary Table 1).

To compare DNA replication timing, we obtained Repli-Seq data across the mouse genome from two germline stages: primordial germ cells (PGC) ($n = 2$, male and female) and spermatogonia stem cells (SSC) ($n = 2$), in addition to 22 other independent mouse cell lines across ten early stages of embryogenesis[29,30]. Repli-seq resolves early and late replicating DNA by labeling with nucleotide analogs at different time points during S-phase followed by high throughput sequencing.[31] We assessed DNA replication time dynamics across cell types by partitioning the mouse genome into 200 kb regions and performing $k$-means clustering of cell-type-specific DNA replication timing data across cell types ($n = 8966$ blocks, Methods). This revealed approximately a third of the mouse genome to be consistently early or late replicating across germline and developmental cell types, where 14% of the genome replicated early and 19% are late replicating (early: RT >0.5, late: RT <−0.5) (Fig. 1C). Enhancers were more prevalent at early DNA replicating regions, however recently evolved enhancers were more likely to be located at consistently late replicating regions compared to conserved enhancers (Conserved: 31.0% early vs. 1.4% late, Recent: 21.2% early vs. 6.3% late) (Fig. 1D). To investigate the emergence of new enhancers, we focused on the averaged replication times across the four germline assays.

The fastest rates of enhancer turnover occurred at late DNA replicating domains. New enhancers were proportionately 1.8 times more common at late than early replicating regions, although the absolute number of enhancers was higher at early replicating domains (Fig. 1E−G and Supplementary Fig. 1). Enhancer turnover was highly correlated with germline replication time ($R^2 = 0.95$), although similar trends were observed comparing using somatic developmental replication time ($R^2 = 0.60$, Fig. 1F and Supplementary Figs. 2, 3). It is unlikely the observed trend is due to an ascertainment bias. Beyond the analysis steps taken to ensure the reproducibility of the peak calls (Methods), we identify the same relationship if we restrict to mouse-specific enhancer chromatin marks at uniquely mappable coordinates that exhibit sequence conservation across species, thereby excluding potential mappability differences which could confound the result (Supplementary Fig. 4).

We next examined poised and active enhancers across four mouse tissues (brain, liver, muscle, and testis). Recently evolved enhancers were consistently later replicating for both poised and active enhancers across the four mouse tissues (Fig. 1H). We found that liver and testis enhancers evolved significantly faster than brain and muscle, with the greatest disparity in enhancer turnover rates between organs at late-replicating regions ($t = 6.77$ and 4.85; $p = 1.2 \times 10^{-07}$ and $3.07 \times 10^{-05}$ for poised and active enhancers, respectively) (Fig. 1I). This result parallels the faster evolution of testis gene expression levels and a slower evolution of brain expression in mammals[32]. When germline replication timing was replaced with somatic timing, the rapid turnover observed in testis remained consistent (Supplementary Fig. 3).

As transposable elements (TEs) are widespread across the genome and have been widely implicated in the turnover of cis-regulatory elements[5,9], we next assessed the relationship between TE evolution and replication timing. We calculated the ratio between the numbers of new TE families and the numbers of ancestral TE subfamilies across replication timing regions. Similar to enhancers, new TE subfamilies of TEs were more abundant in late-replicating regions ($R^2 = 0.96$, Supplementary Fig. 5). Recently evolved enhancers were also more likely to overlap lineage-specific TEs (Fig. 1J). Excluding enhancers overlapping TE (~52% of enhancers) only slightly reduced the slope between enhancer turnover and replication time ($R^2 = 0.94$ and 0.95 for enhancers overlapping and not overlapping TE, respectively) (Supplementary Fig. 6). Hence, the rate of enhancer turnover is not wholly dependent on TE, but both are strongly correlated with DNA replication time across large chromatin domains[9,30,33].

To assess the population of species-specific enhancers that could have emerged from recent copy number duplication events, we clustered lineage-specific enhancers by sequence similarity in human and mouse (Methods). While copy number variants arising from recent homologous recombination events or transposition events are expected to cluster based on shared similarity, sequences emerging from mutations of ancestral sequences should not. The precise degree of inter-enhancer similarity will depend on multiple parameters, including mutation rate, life history, evolutionary pressures, and the evolutionary comparison used for enhancer classification.

Using a significance cut-off of $E = 1 \times 10^{-6}$ and a relaxed sequence coverage threshold of greater than 20% of the query sequence to detect homology among recently evolved enhancers, we find the proportions of singleton enhancers are 75.92% and 77.04% for human and mouse enhancers, respectively. Proportions of singletons were similar between humans and mouse despite a higher number of expected mutations in mouse. As expected, fewer singletons overlapped repetitive elements, including TE, compared to non-singleton enhancers (Fisher's exact test, $p = 3.56 \times 10^{-40}$, odds ratio = 0.13 for human enhancers; Fisher's exact test, $p = 6.8 \times 10^{-171}$, odds ratio = 0.63 for mouse enhancers) (Supplementary Fig. 7). Our results are consistent with the hypothesis that most of these elements did not emerge from recent duplication events.

Enhancer gains are more prevalent in regions that already show enhancer marks or chromatin accessibility in other organs, suggesting the convergent evolution of enhancers due to a rapid pace of evolution[9,34]. To rule out convergent evolution due to rapid turnover, we progressively tightened our conservation criteria. We repeated the analyses, requiring co-occurrence of at least four, and then at least seven, histone marks between the enhancer and other species. We

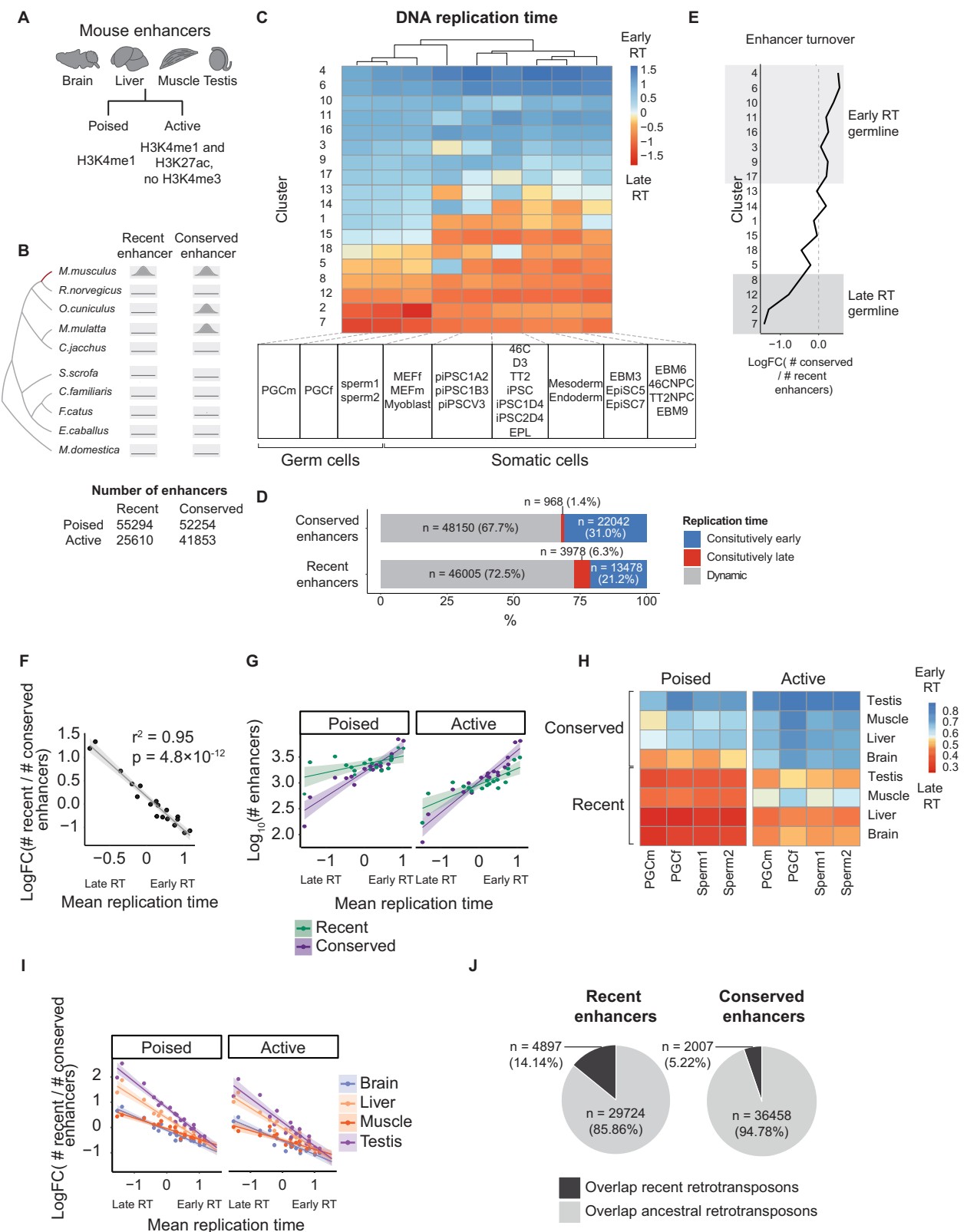

observed highly consistent findings across all these cutoffs (Supplementary Fig. 8). In summary, although most enhancers are found within early germline replicating domains, species-specific turnover was disproportionately enriched in late replication regions. This was the case for both active and poised enhancers. Late germline DNA replication time is associated not only with increased numbers of lineage-specific enhancers but also new subfamilies of TEs (Supplementary Fig. 5). Most

lineage-specific enhancers do not share high degrees of similarities, suggesting the gain of enhancer-associated histone modifications by mechanisms other than duplications.

**Mutations at TF binding sites are linked to enhancer turnover**
TF binding sites can be considered the atomic unit of regulatory element function[35,36]. When mutations occur at TF binding sites, they can

**Fig. 1 | Enhancer turnover is coupled to germline replication timing. A** Mouse enhancers are defined based on combinations of histone marks. **B** Definition of mouse recent and conserved enhancers[9]. Recent enhancers are defined as regions with mouse-specific histone marks enrichment[9]. Conserved enhancers are aligned to regions with regulatory activity in at least two other species. **C** Replication time across 200 kb blocks of the mouse genome ($n = 8966$ blocks) in PGC ($n = 2$ cell lines), SSC cells ($n = 2$ cell lines), and early somatic cell types ($n = 22$ cell lines). Row clustering (blocks) was carried out with $k$-means clustering; columns are cell-type clusters generated with hierarchical clustering. Row clusters were ordered from early (top) to late (bottom) DNA replication timing, across columns (cell-type clusters). **D** Numbers of recent and conserved enhancers in regions of (**C**) with constitutively early (blue), constitutively late (red), and dynamic (gray) replication time. **E** Enhancer turnover as the log fold change of conserved vs. recent enhancers for the 200 kb clusters across mean germline replication time calculated across

PGC ($n = 2$) and SSP cell lines ($n = 2$). Shaded areas represent clusters with constitutive DNA replication time. **F** Scatterplot of mean germline replication time (PGC + SSP) across the 18 clusters shown in (**C**). *P* value from a two-sided test of the Pearson correlation coefficient. Shaded areas represent a 95% confidence interval (CI) of the best fits. **G** Scatterplot of germline mean DNA replication time (PGC + SSP) and $\log_{10}$-transformed numbers of recent and conserved enhancers. Each data point is a cluster defined in (**C**). The shaded region represents the 95% CI of the line of best fit. **H** Mean PGC and SSC DNA replication time of poised and active mouse enhancers separated by tissue and type. **I** Mean germline DNA replication time (PGC + SSP) versus enhancer turnover by tissue and enhancer type. Each data point corresponds to a cluster in (**C**). The shaded region is the 95% CI of the line of best fit. **J** The number of recent/conserved enhancers overlapping recent/ancestral retrotransposons (Fisher's exact test, $p < 2.2 \times 10^{-16}$, two-sided, odds ratio = 2.99).

disrupt or alter the binding of the TF, potentially leading to changes in enhancer activity. Simulation studies have shown new enhancers can evolve within a relatively short evolutionary time due to the accumulation of mutations creating new TF binding sites[12].

We hypothesized that the creation or disruption of TF binding sites could change the activity of enhancers, leading to turnover. As TF binding motifs do not fully explain binding, to test this, we used experimental data from ChIP-seq to train a deep-learning model to predict binding sites. The model takes a 500 bp DNA sequence and outputs a prediction of TF binding based on sequence alone. We expect a higher predicted frequency of TF binding at new enhancers (i.e., those with recently acquired enhancer histone marks) compared to orthologous but non-enhancer sequences.

Our model architecture was trained on human and mouse ChIP-seq of CEBPA and HNF4A in liver. As before, enhancers were defined based on the enrichment of H3K27ac and the absence of H3K4me3 (Methods)[3]. To optimize the learning of shared functional sequences, the model predicts TF binding using a domain adaptive step to remove sequence biases arising from the species-specific genome backgrounds[37]. We retrained this model to ensure enhancer regions used for model testing are excluded from the model training process by removing regions harboring human and mouse-specific enhancers. We then tested human and mouse lineage-specific enhancer sequences to assess whether sequence changes could explain enhancer turnover through their impact on TF binding.

The sequence-based model identified a significantly higher number of HNF4A and CEBPA TF binding sites at human enhancers compared to the mouse orthologs without enhancer marks, suggesting that genetic variation between the sequences is associated with the gain or loss of functional TF binding sites ($p = 5.78 \times 10^{-30}$; OR = 1.95) (Fig. 2A and Supplementary Fig. 9). Conversely, a similar trend was observed when comparing mouse-specific enhancers to orthologous non-enhancer sequences in human ($p = 1.27 \times 10^{-77}$; OR = 3.82) (Fig. 2A and Supplementary Fig. 10). Enhancer turnover was correlated to sequence changes to the canonical binding motifs (Fig. 2B). Moreover, total proportions of species-specific enhancers with predicted HNF4A and CEBPA binding sites were increased at late replicating regions (Fig. 2C). Our findings suggest mutations altering TF binding modulate enhancer chromatin states.

## New enhancers are enriched in eQTLs but lack strong signatures of purifying selection

To understand the selective pressures at enhancers, we used human population variation data to calculate a derived allele frequency (DAF) score in 10 bp windows across the genome using whole-genome sequencing of the relatively isolated Icelandic population (deCODE)[38]. DAF odds ratio (OR) measures the ratio between the numbers of rare and common variants. A high odds ratio indicates an excess of rare variants compared to the background, suggesting purifying selection. We plotted DAF for species-specific and conserved liver enhancers by

centering each enhancer based on functional motifs to increase the power to detect purifying selection (Fig. 3A, B, Methods).

Our results revealed reduced purifying selection acting at recent enhancers relative to evolutionarily conserved enhancers and recent promoters. (Fig. 3C and Supplementary Fig. 11A, B). As expected, a progressive increase in DAF OR at enhancers and promoters was observed with increased degrees of species conservation (Fig. 3D). We note that when the DAF scores significantly differed from genome background, this difference may also be due to a higher frequency of common variants rather than a depletion of rare variants (Supplementary Fig. 11C, D). Such differences can be due to demographic and not selective factors. For example, rare variants may not have had as much time to increase frequency and spread through the population, particularly for recently evolved elements.

The low DAF odds ratios suggest many of these gained ChIP-seq peaks at late-replicating regions may not be as functional in driving gene expression as their early-replicating counterparts. To delve deeper, we tested whether ChIP-seq peaks at late-replicating regions were as likely to activate transcription as early-replicating regions. Using enhancer activity data from human liver enhancers defined by H3K27ac marks tested in HepG2 cells using STARR-seq[39], we compared the normalized activity score between recent and conserved human liver enhancers (recent $n = 254$, conserved $n = 270$). We observed slightly lower activity as measured by MPRA at late replication time, although this was not statistically significant (alpha = 0.05) (Fig. 3E). We further assessed whether enhancers with MPRA activity showed evidence of increased purifying selection compared to tested enhancers without an appreciable level of enhancer activity ("inactive")[39]. MPRA activity did not differentiate constrained enhancers within a species (Fig. 3D). The same trend was observed for recently evolved enhancers, (Fig. 3D). Weak affinity binding could be a potential underlying mechanism for the slight reduction of enhancer activity at late replicating regions[40].

To address whether new and conserved enhancers make qualitatively different contributions to transcription, we tested the relative enrichment of 226,768 significant liver *cis*-eQTLs from healthy individuals in the GTEx consortium (GTEx V7) at human-specific and conserved enhancers. Species-specific enhancers harbored significantly more eQTLs than conserved enhancers (Fisher's exact test, $p < 2 \times 10^{-16}$, odds ratio = 1.2), consistent with an increased frequency of eQTLs at recently evolved promoters in the human genome[41]. That new enhancers harbored more eQTLs than conserved enhancers, would imply recently evolved enhancers are more likely to contribute to standing gene expression variation but may be less important for organismal fitness.

## Tissue-specific evolution of enhancers is linked to late DNA replication timing

Late replicating regions with their dynamically regulated heterochromatin and nucleosome formation potential have been linked to

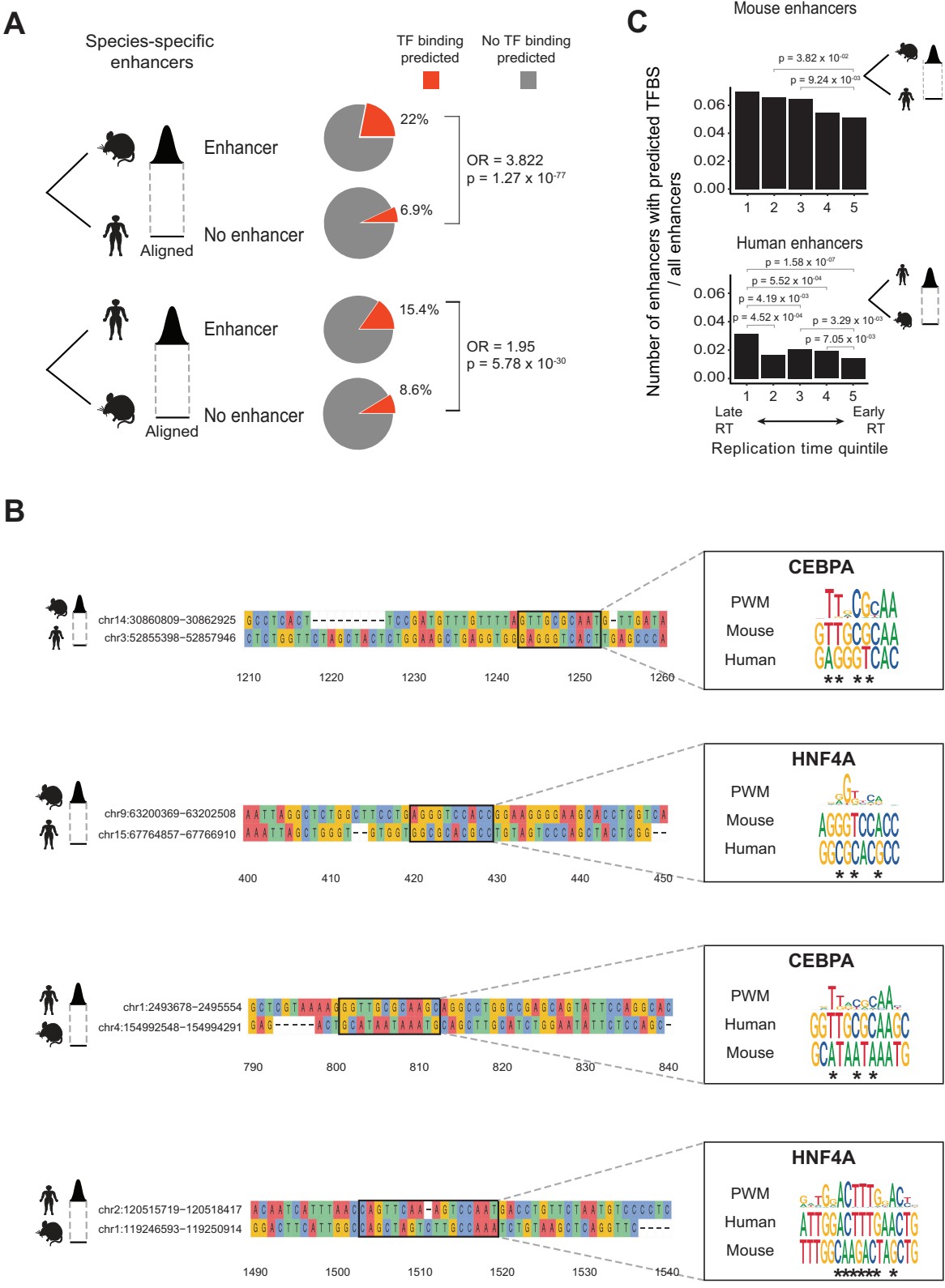

tissue-specific gene expression[24,25,42,43]. Hence, we investigated whether tissue-specific enhancer activity was also associated with late-replicating regions.

Comparing mouse enhancers between the four mouse tissue types, we found tissue-specific enhancers were indeed more likely to be late replicating than enhancers active in more than one tissue (Fisher's exact test, $p = 2.2 \times 10^{-16}$, odds ratio = 0.28, Fig. 4A). Late

replication time is associated with increased tissue specificity regardless of evolutionary age (Fig. 4A). Tissue-specific elements are enriched at late replication time and a faster evolutionary rate than enhancers active in multiple tissues (regression test for difference in slope, $p = 1.27 \times 10^{-5}$; Fig. 4B).

Because tissue-specific control of gene expression is critical during development, we hypothesized that enhancers drive developmental

**Fig. 2 | Deep-learning model links changes in TF binding sites with enhancer turnover. A** Deep-learning domain adaptive model trained with HNF4A and CEBPA binding sites in mouse and human genomes[37]. Prediction on species-specific enhancers and their aligned non-enhancer sequences in the other species. The pie charts show the percentage of enhancers and matched non-enhancer regions with predicted HNF4A and CEBPA TFBSs with a probability threshold ≥ 0.9. Fisher's exact test two-sided *p* values are shown for each enhancer vs non-enhancer comparison. **B** Examples of species-specific liver candidate enhancers and their sequence alignments to the other species where binding is not predicted in (**A**). Boxed alignment of a motif identified in the species possessing the enhancer (top

sequence) and its alignment to the species without the enhancer (bottom sequence). The motif's position-weighted matrix (PWM) logo is on the right. The logo is on the negative strand in the last example. * denotes changes to PWM in the orthologous sequence without peak; Details on the data processing of this figure is available in Supplemental Methods. **C** Numbers of mouse- and human-specific enhancers with predicted TFBSs divided by the total number of enhancers across replication time quintiles. The difference in enhancer proportions was tested using a one-sided Fisher's exact test between all pairs of DNA replication time quintiles, testing for a higher proportion in the latest quintile (alternative = "greater"). *P* values are indicated for significant tests (*p* ≤ 0.05).

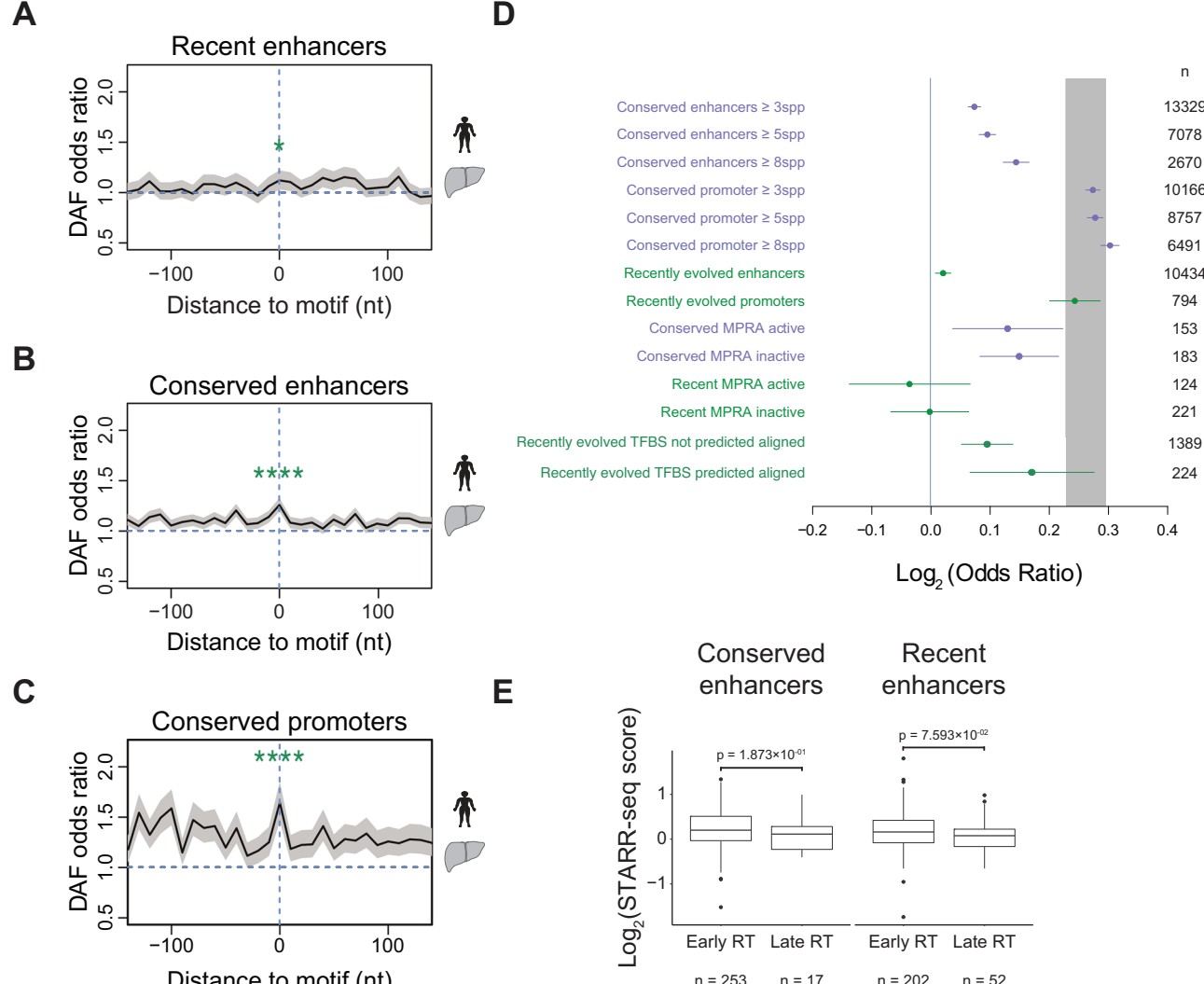

**Fig. 3 | Enhancers do not show strong signatures of purifying selection.** Derived Allele Frequency (DAF) odds ratio for recently evolved (**A**) and conserved human liver enhancers (**B**) and conserved promoters (**C**) compared to background genomic regions as a measure of selection pressure. Promoters and enhancers were centered based on the location of liver-specific functional motifs. *p* = 0.01 and 4.34 × 10⁻¹² for recent and conserved enhancers, and *p* = 8.58 × 10⁻¹⁶ for promoters (two-sided Fisher's exact test, significance code **P* ≤ 0.05 and *****P* ≤ 0.0001). In (**A–C**), the shaded areas represent a 95% confidence interval from sampling the data with replacement (Methods). **D** Log₂-transformed odds ratio of DAF scores for conserved and recent enhancers and promoters. Conservation was defined using multiple thresholds (number of species). Active and inactive enhancers were separated using STARR-seq scores to measure enhancer activity in HepG2 cells[39] (Methods). DAF Log ORs for recently evolved human enhancers aligned to the mouse genome where TFBS were detected or not detected using the deep-learning

model trained for HNF4A and CEBPA in Fig. 2 are shown. The middle points represent the log₂-transformed odds ratio values from a Fisher's test comparing the proportion of rare and common variants against the genome. Error bars represent the 95% confidence intervals of Fisher's exact test. Numbers of elements are shown on the right. **E** Log₂ transformed STARR-seq activity of human liver recent and conserved enhancers separated into early (RT > 0.5) and late (RT < −0.5) replicating. The quartiles in box plots represent the 25th, 50th (median), and 75th percentiles. The interquartile range (IQR) represents the difference between the 75th and 25th percentiles. The upper whiskers extend to the maximum value of data within 1.5 IQR above the 75th percentile. The lower whiskers extend to the minimum value in the data within 1.5 IQR below the 25th percentile. Outliers are values above the upper whiskers or below the lower whiskers. A two-sided Mann–Whitney *U*-test *p* value is shown in each case. The number of enhancers is indicated in each case.

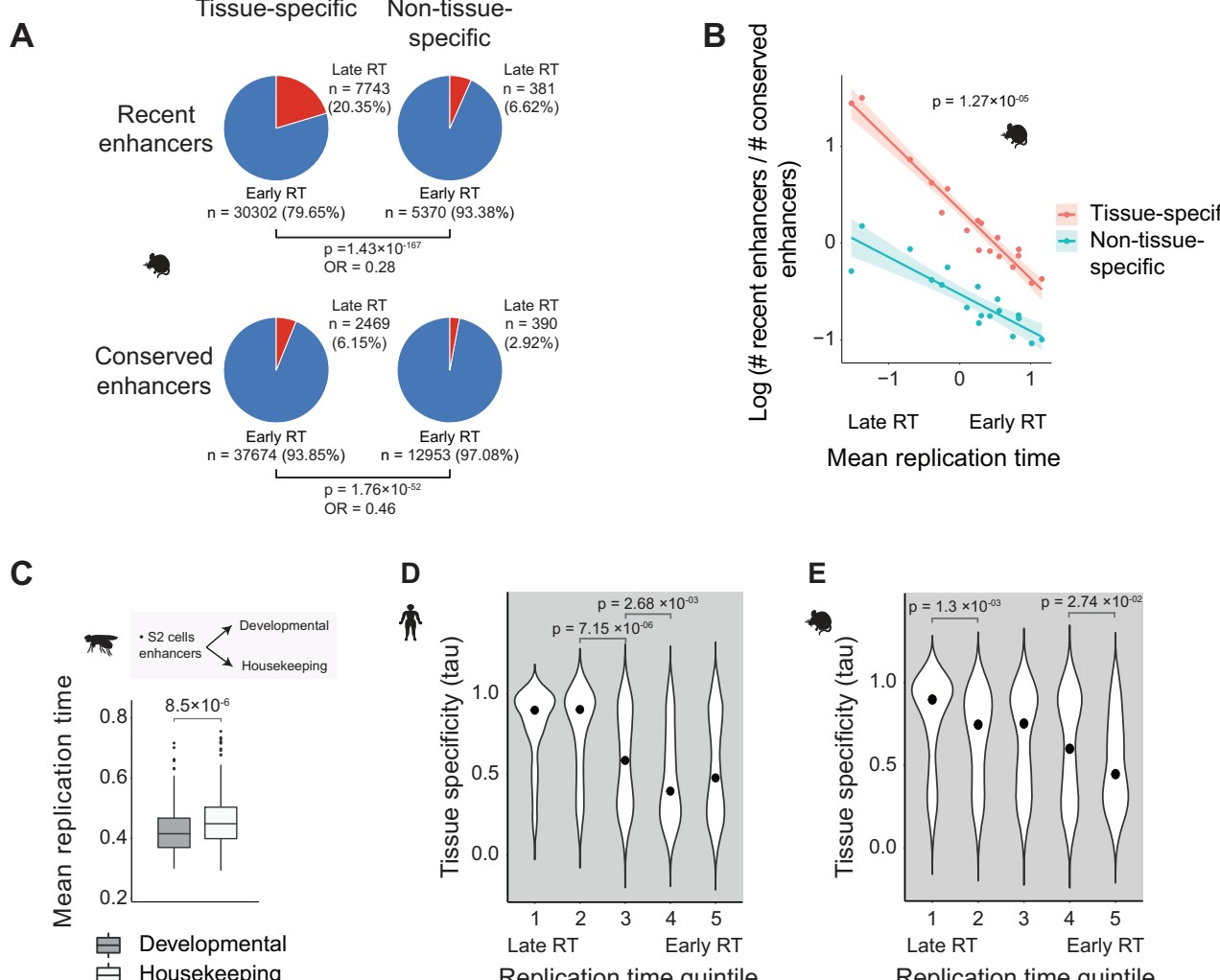

**Fig. 4 | Tissue-specific enhancers are enriched at late replicating regions. A** The proportions of early and late replicating enhancers for tissue-specific and non-tissue-specific mouse recent and conserved enhancers (defined with four tissues: brain, liver, muscle, testis) (Two-sided Fisher's exact test, *p*, and odds ratio values are shown in each case). **B** Mean mouse germline DNA replication time versus enhancer turnover rate, defined as log (number of recent enhancers/number of conserved enhancers), for tissue-specific and non-tissue-specific enhancers (shown in red and blue, respectively) across the 18 DNA replication time clusters shown in Fig. 2C. $R^2 = 0.95$ (two-sided Pearson correlation *p* value = $6.43 \times 10^{-12}$) and 0.78 (two-sided Pearson correlation *p* value = $1.19 \times 10^{-06}$) for tissue-specific and non-tissue-specific enhancers, respectively. ANCOVA *p* value for the difference in slope is shown. The shaded area represents a 95% confidence interval of the best fit. **C** Mean replication time of developmental and housekeeping fruit fly enhancers (one-sided Mann−Whitney *U*-test housekeeping versus developmental,

alternative = "greater," *n* = 200 enhancers each class). The quartiles represent the 25th, 50th (median), and 75th percentiles. The interquartile range (IQR) represents the difference between the 75th and 25th percentiles. The upper whiskers extend to the maximum value of data within 1.5 IQR above the 75th percentile, and the lower whiskers extend to the minimum value in the data within 1.5 IQR below the 25th percentile. Outliers are values above the upper whiskers or below the lower whiskers. **D, E** Violin plots of tissue-specific expression scores (tau values) of human and mouse TFs separated into five quintiles depending on their respective motif enrichments at early versus late replicating enhancers (one-sided Mann−Whitney *U*-test, pairwise comparison of later vs. earlier replicating quintile, alternative = "greater," significance code: 'ns' *P* > 0.05, \**P* ≤ 0.05, \*\**P* ≤ 0.01, and \*\*\*\**P* ≤ 0.0001). *P* values for the significant (*p* ≤ 0.05) comparisons of consecutive quintiles are shown. Across the panel, mouse germline replication times are calculated as the mean across PGC and SSC cells, and human replication times are from H9 cells.

expression may replicate later than those that drive housekeeping expression. Indeed, enhancers associated with developmental promoter activity in *Drosophila* were later replicating than enhancers associated with the housekeeping promoter (Fig. 4C; Mann−Whitney *U*-test, *p* = $8.5 \times 10^{-6}$; Methods). This pattern was consistent across different chromosomes independent of the promoters' endogenous location (Supplementary Fig. 12).

We then examined whether enhancers located at late replicating regions were also associated with binding more tissue-specific TFs. Using an established index of tissue-specificity of gene expression, tau[44], we examined tissue-specific TF expression for 477 and 360 genes in humans and mice across 27 and 19 tissues, respectively[45]. TFs were partitioned into five groups based on the relative enrichment of their motifs at

enhancers from early and late replication time. TFs whose motifs were most enriched motifs at late replicating enhancers were significantly more likely to show tissue-specific expression patterns (Fig. 4D, E).

## Developmentally associated motifs were enriched at late DNA replication time

In mammals and other warm-blooded vertebrates, DNA replication time is also linked to long regional stretches of compositionally homogeneous DNA with uniform GC base composition[26,27,46–49]. These are known as GC isochores and are distinct between early and late replicating regions[48,50,51]. The origin of isochores can be partially explained by mutational biases[2,26,27]. Late replicating sequences harbor a biased substitution pattern towards A and T nucleotides[21,52,53], where

the primary contributor is the deamination of methyl-cytosine at CpG sites, resulting in C > T transitions.

Given the nucleotide differences in TF binding sites, we sought to understand how the GC isochore may impact genome-wide TF binding dynamics. GC isochores are closely correlated to replication timing. We confirmed that the loss of G and C nucleotides was greatest at late DNA replication time using base substitutions inferred from the last common ancestor of *Homo* and *Pan* (Fig. 5A). Beyond TF binding sites, we found the relationship between GC content and DNA replication time was also broadly reflected at cis-regulatory elements (Fig. 5B) levels, but enhancers contained higher GC than genomic background.

GC isochores corresponded to a profound shift in the counts of different TF binding motifs at enhancers across replication times (Fig. 5C, D and Supplementary Fig. 13). In humans and mice, the most prevalent motifs at late replicating enhancers were AT-rich, while early replicating enhancers were GC-rich (Fig. 5C, D). Homeodomain factor motifs, which act as critical regulators in development, were predominantly enriched in late-replicating enhancers (Fig. 5E and Supplementary Fig. 14). A similar trend was observed at promoters and in mouse (Supplementary Figs. 15, 16). Restricting to the top-scoring motifs resembling the consensus homeodomain TFBSs did not change the observed trend (Fig. 5E). To interrogate this further, we also compared TF motif enrichments at regions randomly sampled from the genome (Fig. 5E). Motif enrichment was well predicted by replication time, but replication timing did not explain all enriched motifs (e.g., HOXC13, HNF1B, and POU4F3) (Supplementary Fig. 16). Some motifs possess a different nucleotide composition than predicted by replication time, suggestive of natural selection.

Because the nucleotide frequencies of motifs at enhancers are an indirect measure of TF binding, we asked whether the observed trends are reflected in vivo. Using the DNA binding locations of 71 proteins from ChIP-seq data in human K562 cells, we found TF binding sites were bimodally distributed with respect to replication timing (Supplementary Fig. 17). We fitted Gaussian components using mixture modeling for each protein, focusing on binding sites at later replicating time, which were more variable between TFs than early replicating TFBS (Supplementary Fig. 17A, B and Supplementary Data 1). Our results suggest that DNA replication time impacts the type and frequency of TF binding motifs, thereby influencing the TFs bound at these regions (Supplementary Fig. 17C). However, the in vivo pattern is attenuated compared to motifs identified by computational search reflective of the complex regulatory interplay between DNA sequence and other factors, including epigenetic modifications, protein cooperativity, and the topological chromatin context.

In summary, DNA replication time is associated with not only the tempo/rate of enhancer evolution but can also influence the type of TF motifs that are enriched due to its association with large-scale shifts in GC content, with the potential to impact which TFs are recruited.

### Enhancer turnover in cancer is enriched at late DNA replication time

Finally, we investigated whether the observed link between DNA replication timing and cis-regulatory element turnover is conserved across evolutionary timescales. We analyzed enhancer gains and losses across four cancer types relative to DNA replication time (Fig. 6A). Prior studies have shown DNA replication time is maintained in cancer, allowing for comparisons with healthy samples[54,55]. We defined a "gain" of enhancers as those characterized in cancer cell lines but not in the non-diseased state. Inversely, enhancers in the healthy cell state but not in cancer were defined as "lost." Enhancers annotated in both states were termed 'unchanged' (Fig. 6A, B and Supplementary Table 2).

We annotated candidate enhancers in healthy breast, prostate, thyroid, and pre-leukemic cells and in their diseased states[56–60]. In breast and prostate cancer, enhancers were defined by ChIP-seq of histone marks (Fig. 6A, Supplementary Fig. 18A, and Supplementary

Table 2). In AML and thyroid cancer, enhancers were defined as distal chromatin accessible in patient-matched primary tissues and tumors (thyroid cancer and matched healthy $n = 3$, pre-leukemic and matched blast cells $n = 3$). We used DNA replication time information for prostate cancer cell line (LNCaP), healthy prostate epithelial cells (PrEC), and breast cancer cell line (MCF-7)[54], with predicted DNA replication time in pre-leukemic and thyroid cells (Methods).

Consistent with cross-species results, we found the highest rate of enhancer turnover at late DNA replicating domains compared to enhancers that remained unchanged (Fig. 6C–F and Supplementary Fig. 18B–G). This trend was unaffected by differences in recombination breakpoints[61] (Supplementary Fig. 19). Subsequently, we compared cancer variants at gained, lost, and unchanged enhancers in thyroid, AML, and prostate cancer. We used matched tumor and healthy samples from the same individual to calculate somatic mutations due to cancer. Prostate cancer variants were identified from the whole-genome sequencing of the prostate cancer genome cell line, and common population variants were removed to focus the analyses on somatic mutations only[62].

Mutation numbers were elevated for enhancers gained or lost compared to unchanged enhancers across the three cancer types for which we had variant data (Fig. 6G–I, Supplementary Fig. 18H–N, and Supplementary Table 2). The trend was consistent across all individuals for thyroid cancer and AML (Supplementary Fig. 20). Our results demonstrate that in cancer, as well as across evolution, a higher turnover of cis-regulatory elements occurs at late DNA replication time. This turnover is associated with increased mutational burden, suggesting that mutations could play a causal role in the emergence or loss of enhancers in tumorigenesis.

## Discussion

In this study, we demonstrated the significance of genome structure on enhancer evolution. While most enhancers, defined by histone mark occupancy, were identified in early replicating regions, comparative analyses showed that young enhancers were almost twice as likely to replicate later than conserved enhancers. Genetic changes during evolution can create or abolish TF binding sites associated with the emergence of cis-regulatory elements or decommissioning of existing elements. The short length of TF binding motifs and their degeneracy allows for the rapid emergence and fixation of TF binding motifs[11–16]. We found that enhancer turnover is linked to sequence changes that alter TF binding. Remarkably, similar patterns in cis-regulatory evolution were evident in mammalian evolution, spanning millions of years, and in cancer cells, occurring over months or years, suggesting that regulatory evolution is intertwined with the evolution of genome architecture across time scales.

Our definition of species-specific enhancers depended on the other species in the comparison. The closest relatives to humans and mice used in our comparisons were macaque and rats, respectively, with divergence times of ~29 (human vs. macaque) and ~12 (mouse vs. rat) million years ago[63]. Based on the mutation rate and generation time for each species, we should expect ~15 mutations per kb in humans and ~120 mutations per kb in mice since the last common ancestor of human/macaque and mouse/rat, respectively (based on mouse mutation rate of $5 \times 10^{-9}$ per base per generation, human mutational rate of $1.28 \times 10^{-8}$, generation time of 0.5 and 25 years[64,65]). Hence, recently evolved mouse enhancers are expected to exhibit greater inter-species variation compared to human-specific enhancers.

Our results imply that evolutionary innovation in gene regulatory modules is more likely to emerge from regulatory elements at later replication domains in a tissue-restricted manner. Indeed, disparities in enhancer turnover between organs were also most pronounced in late-replicating regions. Late replicating regions are associated with developmental enhancer activity and are enriched for developmentally relevant TF binding sites related to body patterning (e.g.,

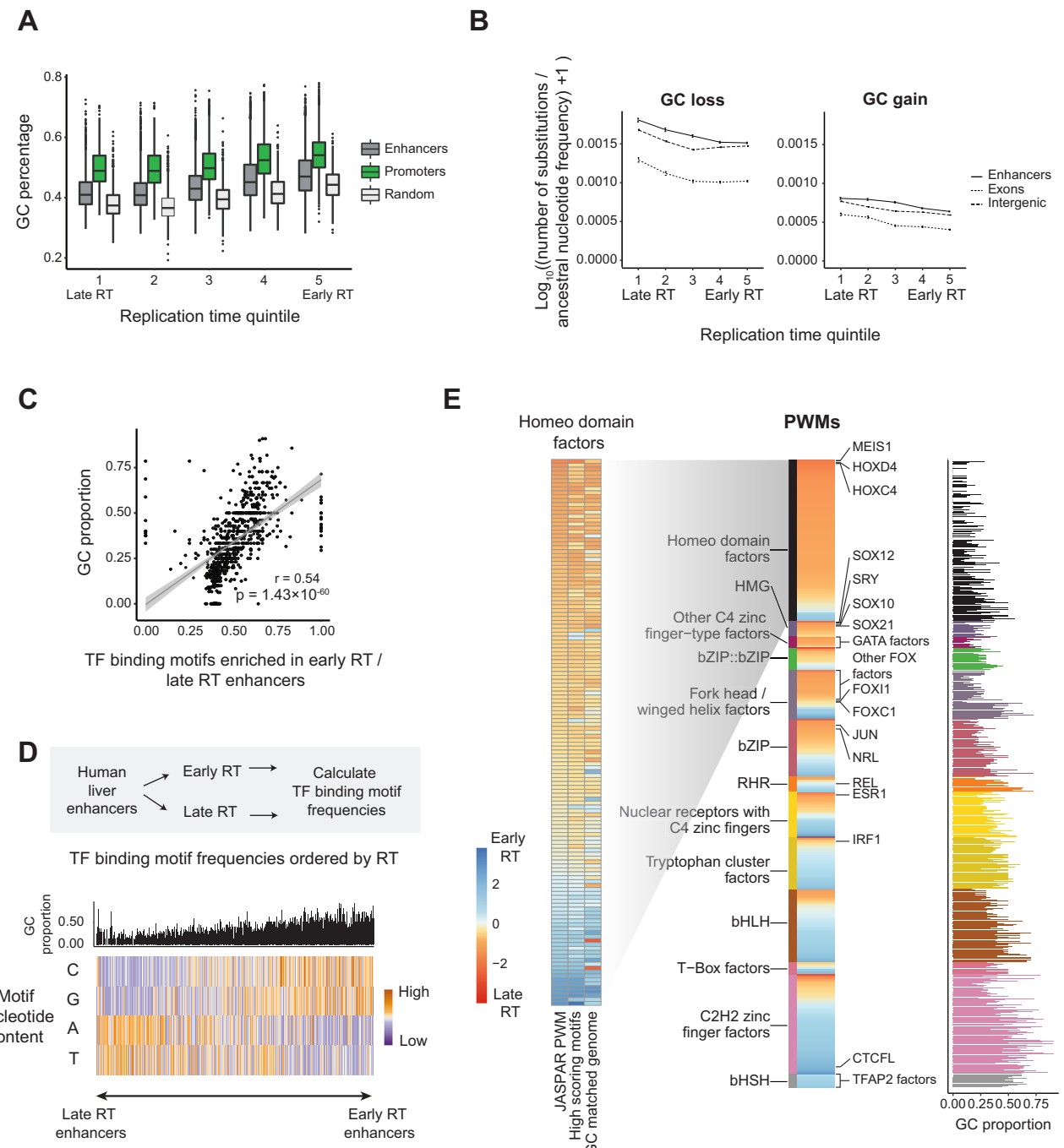

**Fig. 5 | AT-rich motifs are associated with developmental TFs and are over-represented at late replication time in mammals. A** GC percentage of human liver enhancers and promoters and random genomic regions across replication time quintiles (random regions were sampled from the non-genic areas of the genome, excluding promoters and enhancers) (n = 28,175, 11,520, and 5000 enhancers, promoters, and genomic background, respectively). Difference in GC% across DNA replication time quintiles was significant for every type of sequence (Supplementary Table 3). Quartiles, whiskers, and outliers are defined in Fig. 3E. **B** Mean non-CpG substitutions at liver enhancers, exonic, and intergenic regions across H9 replication time quintiles. Substitutions calculated between humans and the inferred common ancestor of *Homo* and *Pan*. The number of substitutions was adjusted by their ancestral nucleotide frequency, and $\log_{10}$ transformed. Error bars represent standard error (the number of regions per quintile is shown in Supplementary Table 4). **C** Scatterplot of the proportion of GC for TF binding motifs based on enrichment at early versus late replicating human liver enhancers. Two-sided

Pearson correlation coefficient and *p* value are shown. The shaded area represents the 95% confidence interval of the best fit. **D** The bar plot shows the GC proportion of each motif. Heatmap of the GC/AT nucleotide content of TF binding motifs ordered based on their relative enrichment at early versus late replicating human liver enhancers (n = 5538 each replication time). Each column shows a human TF binding motif from the JASPAR database. **E** Relative enrichment of TF binding motifs at early versus late replicating liver enhancers grouped by TF class (center heatmap). The GC content of the motifs is shown on the right. Bars are colored by TF Class. Only TF classes with more than ten TFs are shown. The heatmap on the left shows the relative enrichment of homeodomain factors in early versus late replicating enhancers using JASPAR human motifs (left column) and using only the highest scoring motifs (mid column) (Methods). The column on the right shows the relative enrichment of homeodomain factors in early versus late GC%-matched genome background.

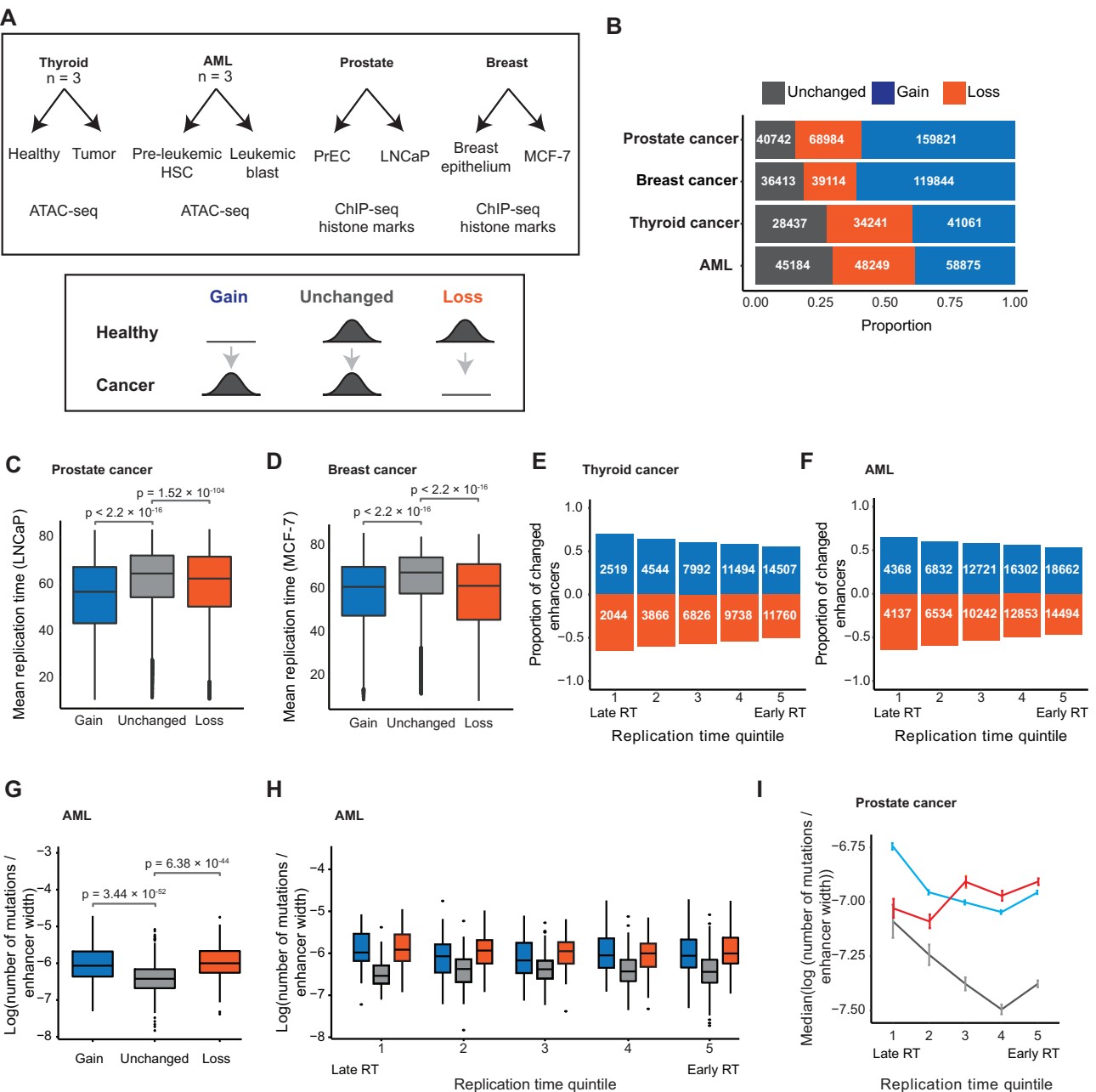

**Fig. 6 | Enhancer turnover is enriched at late replication time in cancer.**
**A** Overview of the enhancer datasets in cancer and matched healthy tissues and cell lines (top). Gained, unchanged, and lost enhancers were defined in each cancer type (bottom). **B** Proportions of unchanged, gained, and lost enhancers in each cancer type. **C, D** Replication time of gains, unchanged enhancers, and losses in the prostate (**C**) and breast (**D**) cancer (two-sided Mann–Whitney U-test; p value is shown for each comparison). A similar trend exists in thyroid cancer and AML (Supplementary Fig. 18B, C). The quartiles represent the 25th, 50th (median), and 75th percentiles. The interquartile range (IQR) is the difference between the 75th and 25th percentiles. The upper whiskers extend to the maximum value of data within 1.5 IQR above the 75th percentile, and the lower whiskers extend to the minimum value in the data within 1.5 IQR below the 25th percentile. Outliers are values above the upper whiskers or below the lower whiskers. **E, F** Proportions of enhancer gains and losses in thyroid cancer (**E**) and AML (**F**) are relative to the

number of unchanged enhancers across replication time quintiles. Proportions of losses were multiplied by (−1). **G** Log transformed the number of mutations normalized by enhancer width for AML gains, losses, and unchanged enhancers (two-sided Mann–Whitney U-test; p value is shown for each comparison). The boxplot quartiles and outliers were defined as in (**C, D**). **H** Log transformed the number of mutations normalized by enhancer width in AML across replication time quintiles (two-sided Mann–Whitney U-test). The boxplot quartiles and outlier values were defined as in (**C, D**). **I** Median log-transformed number of mutations normalized by enhancer width at prostate cancer gains, unchanged enhancers, and losses across replication time quintiles (error bars represent standard error). The numbers of enhancers and mutations in each cancer type are in Supplementary Table 2. Cell type-specific replication timing datasets are used (Methods). The numbers of enhancers in each group in panels **C, D, G–I** are shown in Supplementary Table 5.

homeobox). Consistent with this, tissue-specific enhancers are enriched at late replicating regions, where they are associated with tissue-specific TFs.

Our findings also provide a compelling explanation for tissue-specific gene expression differences in mammals linked to GC isochores

whose position mirrors replication domains[54,66]. We showed that the highly organized isochore patterns in mammalian genomes influenced the genome location and frequencies of TF binding sites, with specific types of TFs more likely to be recruited at certain replication timing domains. Specifically, tissue-specific TFs are frequently linked to AT-rich,

late replicating binding sites, which may explain the observed enrichment of tissue-specific gene expression in GC-depleted regions[24,25].

We speculate that transcriptional changes occurring at late replicating domains may have played a pivotal role in the evolution of the bilaterian body plan and embryonic development of multicellular organisms. Notably, replication timing is dynamic between cell types and varies between germline and somatic cell types[29,67,68]. Approximately 30% of the human genome switches between replication timing domains across 26 human cell lines[69]. Therefore, enhancers emerging in germ cells at late replication time could shift to earlier replicating domains in differentiated cell types, where they may have a significant influence on gene activity.

## Methods

### Mammalian enhancer annotation

Unless specified otherwise, all analyses were performed on the human and mouse genome assemblies hg19 and mm10. R v4.0.0[70] was used. Species-specific ChIP-seq datasets were defined using multi-species ChIP-seq datasets[3,9]. To summarize, the candidate enhancer identification strategy reads were aligned using BWA v.0.5.9/0.7.12, and peaks were called using MACS v.1.4.2/2.1.1 using total DNA input control with $p < 1 \times 10^{-5}$ threshold. Consensus peaks that overlapped two or three biological replicates by a minimum of 50% length were used. Enhancers were defined as those regions that overlapped an H3K27ac or H3K4me1 enriched region but not a H3K4me3 enriched region.

Conserved human enhancers ($n = 13{,}329$) were defined as liver enhancers in at least two of 18 other mammalian species (Rhesus macaque, green monkey, common marmoset, mouse, rat, Guinea pig, rabbit, Northern tree shrew, dolphin, sei whale, Sowerby's beaked whale, cow, pig, dog, cat, ferret, opossum, and Tasmanian devil)[3]. Recently evolved (i.e., human-specific) enhancers ($n = 10{,}434$) were defined as human cis-regulatory elements without a histone mark indicative of enhancer activity in another species at aligned regions (~85%) or did not align to the genomes of other species (~15%). Supplemental Table 6 and Supplementary Data 2 show the mean enhancer width per dataset (human and mouse) and the number of human enhancers aligned to other species' genomes, respectively.

The alignment of mouse recent enhancers to other species' genomes was determined based on liftOver mapping with option -minMatch = 0.6. We used chain files for the assemblies RheMac10, CalJac4, Rn6, OryCun2, SusScr11, CanFam3, FelCat9, EquCab3, and MonDom5.

To check the overlap of conserved and species-specific enhancer with chromatin-accessible regions, we used ATAC-seq data for the human liver and DNAse-seq data for the mouse brain, liver, and muscle (Supplementary Table 1). The minimum overlap of an enhancer with ATAC-seq or DNAse-seq peaks was 30% of the enhancer base pairs in all cases. The alternative hypothesis tested was a higher overlap of conserved enhancers with accessible regions using the option alternative = "greater" in Fisher's exact test. Conserved enhancers were conserved in at least two other species.

### Replication time data

Repli-seq data was generated by treating cells with 5-bromo-2'-deoxyuridine (BrdU), a thymidine analog, to label newly synthesized DNA. Subsequently, cells are fixed and FACS-sorted based on their DNA content into early S-phase and late S-phase cell populations. The DNA from these cells is then amplified and mapped to the reference genome. To quantify the timing of DNA replication, the ratio of normalized read coverage between the early and late fractions is calculated[71]. Higher values in this ratio represent early DNA replication; low values indicate late replication. The replication time of every enhancer was calculated by averaging the replication time of the regions they overlap. Early and late replication elements were denoted as mean times >0.5 and <−0.5, respectively. The difference in the probability of recent enhancers between the early and late replication time was calculated as follows:

$$\frac{P(Recent\ enhancer | Late\ RT)}{P(Recent\ enhancer | Early\ RT)} \qquad (1)$$

Z-score transformed replication timing data was obtained from human ESC H9[72], mouse primordial germ cells (PGC), spermatogonial stem cells (SSC)[30], and 22 mouse cell lines differentiated from ES cells[29]. Mean mouse germline replication time was calculated across PGC ($n = 2$) and SSP cell lines ($n = 2$). Mean somatic replication time was calculated across all 22 mouse cell lines. For cancer analyses, DNA replication time information for prostate cancer cell line (LNCaP), healthy prostate epithelial cells (PrEC), and breast cancer cell line (MCF-7)[54], together with predicted pre-leukemic and thyroid DNA replication time data using ATAC-seq information was used (see below). All genomic regions with available replication time data were included in downstream analyses.

### STARR-seq data for HepG2

We used STARR-seq data of human liver enhancers defined by ChIP-seq and tested on the HepG2 cell line[39]. After removing negative controls, we separated the tiles ($n = 6735$) into active and inactive groups using the published threshold ($\log_2$ score >1) and overlapped with our enhancers.

### Estimation of DNA replication time using ATAC-seq

Where relevant replication timing data was unavailable, ATAC-seq data was used to infer replication time using Replicon v0.9[73]. ATAC-seq signal was normalized to a mean of 0 and unit variance. Replicon was run with default options on every chromosome (excluding scaffolds). The predicted replication time values were multiplied by −1 to match the direction of Repli-Seq RT values. The mean ATAC-seq signal across pre-leukemic samples was used to predict replication time.

### Clustering of mouse replication time data

Replication time data from 22 early embryonic mouse cell lines differentiated from mouse embryonic stem cells were transferred to mm10 coordinates (USCS liftOver tool) and overlapped with the replication time regions from PGC ($n = 2$ cell lines) and SSP cell lines ($n = 2$ cell lines)[29,30]. Mean replication time was calculated for every 200 kb region across the mouse genome ($n = 8966$ replication time bins across all cell types). Mean replication time values were centered and scaled using the function "scale"[70]. The function "Mclust" from the R package mclust was used to estimate the best number of $k$-means clusters of 200 kb replication time bins (G = 1:k.max, modelNames = mclust.options ("emModelNames"), where k.max is 20) (mclust version 5.4.6)[74]. The best number of clusters was selected based on its Bayesian Information Criterion (BIC) (k = 18, BIC = −37291.04). Replication time bins clusters were obtained with the function kmeans (18, iter.max = 20). Cell types were clustered using hierarchical clustering with k = 9 (function hclust from base R, method = "complete", distance = "Euclidean").

### Tissue-specificity score

Tau ($\tau$) scores of tissue specificity were calculated with the following formula:

$$\tau = \frac{\sum_{i=1}^{N}(1 - x_i)}{N - 1} \qquad (2)$$

Where $N$ represents the number of tissues, $x_i$ represents the expression profile of one tissue divided by the maximum expression value across tissues[44]. This analysis used the previously described mouse tissue data from the brain, liver, muscle, and testis.

## Model to test for tissue-specific differences

We used a linear model to test for differences in tissue-type-specific evolutionary rates. Using the formula logFC_enh ~ mean_RT + tissue_pair + tissue_pair:mean_RT, where logFC_enh represents the log (number of recent enhancers/number of conserved enhancers) values, tissue_pair is the tissue pair codified as binary (liver and testis = 1, brain and muscle = 0) and mean_RT is the mean germline replication time. The interaction effect was tested using the *t*-test statistic.

## Transposable elements

Transposable elements from RepBase (v27.04 for mouse and human)[75] were annotated using RepeatMasker (v4.0.6 for mouse, v4.1.0 for human) using the sensitive search setting for mouse ("-s")[76]. Species-specific elements, as classified by RepBase, were termed "recent," and the remaining TEs termed "ancestral."

## Developmental enhancer analysis

Summit coordinates of *Drosophila* enhancers determined using STARR-seq on the S2 cell line were downloaded[77]. Housekeeping and developmental promoter were of *RpS12* and *even-skipped* TF, respectively. We defined housekeeping and developmental enhancers as the most highly ranked 200 enhancers for each promoter based on their STARR-seq score. Fly DNA replication time profiles for S-phase in S2-DRSC cells were used[78]. Enhancer summits were extended by 250 bp upstream and downstream, and each enhancer's mean replication time was calculated.

## Cross-species deep-learning model

We used the model architecture from ref. 37 and retrained their model to ensure our test data was not used in the training process and to focus the model on learning differences at orthologous regions that show species-specific histone marks indicative of differences in enhancer activity between human and mouse genomes.

A domain adaptive neural network architecture was used to remove background sequence biases between human and mouse genomes at TF binding sites[37]. Input data were generated by splitting the mouse (mm10) and human (hg38) genomes into 500 bp windows, with 50 bp offset. After excluding all regions containing human- and mouse-specific enhancers and their orthologous region in the other species, we trained the model described in ref. 37. Liver human and mouse HNF4a and CEBPA ChIP-seq peak data from[2] were remapped to hg38 or mm10, respectively. We trained two sets of models for every TF: humans as the source species and mice as the source species. For each species, peaks were converted to binary labels for each window in the genome: "bound" (1) if any peak center fell within the window, "unbound" (0) otherwise. We constructed balanced datasets for training using all bound regions and an equal number of randomly unbound samples (without replacement). Sequence data was one hot encoded. Human and mouse genome sequences were used for model training, excluding Chr 1 and Chr 2. Genome windows from Chr 2 were used for testing. Genome windows from Chr 1 were used for validation. For each TF and species, models were trained for 15 epochs to reduce bias (Supplementary Fig. 21). Final models were selected based on maximal auPRCs. The test dataset comprised species-specific enhancers centered on the middle 500 bp of each element. For predictions using the model, we used a probability ≥0.9. USCS "liftOver" with minMatch = 0.6 was used for genome assembly remapping. We selected models that maximized the auPRC. We evaluated the performance of the models using test datasets (Supplementary Fig. 22). We used the models to predict TF binding in species-specific enhancers centered on the middle 500 bp of each element.

## Natural selection analysis

Human genome variation data were retrieved from the deCODE whole-genome sequencing study of the Icelandic population[38]. Derived allele frequency (DAF scores) of every segregating SNP was calculated, and alleles were defined as either rare (<1.5% population frequency) or common (>5% frequency) as previously described[41]. The number of rare and common alleles in 10 bp windows were centered with respect to the locations of functional liver-specific TF binding motifs from the database funMotifs (v1.0), and 75 unique motifs were annotated[79]. These counts were normalized for the average rates with 2–4 kb upstream and downstream flanking regions. Confidence intervals were obtained by performing 100 bootstrap replicates of sampling the motif locations with replacement. Odds ratios of rare against common alleles between enhancers (and promoters) and size-matched background genomic regions selected randomly were calculated in 10 bp windows. Odds ratio confidence intervals and *p* values were obtained using Fisher's exact test. Only autosomes were considered.

## Copy number analyses

Homology was assessed using blastn with the option -max_target_seqs N (blast + /2.11.0)[80]; this option was used to retrieve the maximum number of hits for every enhancer; N represents the number of enhancers in every dataset, 10,434 and 80,904 for human and mouse, respectively. Hits were filtered by E-value <1 × 10⁻⁶ and query coverage >20. We defined singleton enhancers as enhancers without significant similarity to other enhancers.

## Motif frequency analysis

Motif enrichment in human and mouse enhancers and human promoters used the function annotatePeaks.pl from HOMER (v.11) (option -size given) with human motifs from the JASPAR 2020 database (*n* = 810). The reference genome annotation was provided through the option -gtf[81]. To calculate the nucleotide composition of JASPAR motifs, a nucleotide was assigned to a position of the PWM matrix if its frequency was higher than 0.5. Otherwise, an "N" is set to that position. The proportion of every nucleotide is calculated with respect to the length of the motif (number of bases).

Motif replication time was calculated as the relative enrichment of a motif early against late replicating enhancers. For each motif, we used the formula $\log_2$ ((present_early/absent_early)/(present_late/absent_late)), where present_early is the number of early replicating enhancers (RT >0.5) with non-zero motif instances of a given motif and absent_early is the number of enhancers with zero motif instances. Similarly, present_late and absent_late represent the number of late replicating enhancers (RT <−0.5) with non-zero and zero motif instances, respectively. This measure reflects the relative abundance of the motif between early versus late replicating enhancers. The nucleotide composition of motifs was calculated based on the motif consensus sequence. High-scoring homeodomain motifs were defined as the motifs in the top quintile of all human homeodomain motifs according to their score from HOMER annotatePeaks.pl.

We identified genomic regions with matched GC content using the function genNullSeqs from the R package "gkmSVM" (version 0.83.0). GC%, sequence length, and repeat content were matched with 2% tolerance (repeat_match_tol = 0.02, GC_match_tol = 0.02, and length_match_tol = 0.02, batch size = 5000, nMaxTrials = 50).

## Gaussian mixture models

We collected Chip-seq data for the binding sites of 71 transcription factors. Hg19 coordinates were used. We overlapped all the TF binding sites with H9 ESC DNA replication time and calculated each TF binding site's mean DNA replication time. Afterward, we built a Gaussian mixture model for every TF using the function normalmixEM with k = 2 to get two components (R package mixtools version 1.2.0)[82]. The function returns mu, sigma, and lambda values for each component. Mu represents the mean DNA replication time; sigma denotes the standard deviation; lambda indicates the final mixing proportions (i.e., the contribution of each component to the final mixture distribution).

## Cancer datasets

ChIP-seq data from the prostate cancer cell line, LNCaP, was used to annotate enhancers[59]. For healthy prostate epithelial cells (PrEC), enhancers were defined using chromHMM[60]. We used ChIP-seq data of histone marks in the breast cancer cell line, MCF-7, to annotate enhancers and healthy epithelial breast cells (patient's epithelium samples). LNCaP, MCF-7, and breast epithelium enhancers were defined as enriched in H3K27ac or H3K4me1, excluding proximal regions (±1 kb from TSS).

## Cancer ATAC-seq pre-processing and peak calling

We used matched ATAC-seq from cancer and healthy thyroid samples from three individuals[57], and ATAC-seq files for the matched pre-leukemic and blast cells from three individuals[56]. ATAC-seq fastq files for the matched cancer and healthy thyroid samples from three randomly chosen individuals were downloaded[57]. Adapter sequences were identified and removed using BBDuk (ktrim = r k = 23 mink = 11 hdist = 1 tpe tbo, http://jgi.doe.gov/data-and-tools/bb-tools/). Trimmed reads for each sequencing run were mapped to genome assembly hg19 with bowtie2 v.2.3.5.1 in paired-end mode[83]. Discordant and poor-quality reads were removed (-f2 -q30 -b), and the output was sorted with samtools v.1.10[84]. The obtained.bam files were merged by sample (MergeSamFiles) with duplicates removed (MarkDuplicates, http://broadinstitute.github.io/picard/), resulting in three tumors and three healthy libraries. ATAC-seq fastq files for the matched pre-leukemic and blast cells from three randomly chosen individuals with AML[56] were processed similarly. For each sample from each cancer type, peaks were called using MACS[85] (-g hs -f BAMPE -B and default q-value cutoff of 0.05), and a union set of peaks was defined.

## Cancer variant calling

We restricted our mutational analysis to single nucleotide polymorphisms (SNPs). Prior to variant calling in thyroid cancer and AML enhancers, we corrected for systematic bias and other sequencing artifacts. Base quality scores of ATAC-seq reads were recalibrated with BaseRecalibrator and ApplyBQSR (GATK v4.2.5.0[86]) using variants from 1000 Genomes and the Database for Genomic Variants (--known-sites Mills_and_1000G_gold_standard.indels.b37.sites.vcf --known-sites Homo_sapiens_assembly19.known_indels_20120518.vcf --known-sites dbsnp_138.b37.vcf.gz). To distinguish somatic mutations, in addition to Genome Aggregation Database (gnomAD), we generated a custom database from the matched healthy samples to filter out the patient-unique germline variants. A panel of healthy (pon) was developed by calling variants on healthy samples in Mutect2[87] (with option --max-mnp-distance 0), at open chromatin regions identified by MACS3. FilterMutectCalls was used to remove population-level standing variation, patient-specific germline variants, and variants that show alignment, strand, or orientation biases. This was done by selecting variants marked by PASS in the FILTER field of the result.

For prostate cancer, variants called from the whole genome sequence of LNCaP[62] were mapped from hg38 to hg19 using Picard LiftoverVcf (v2.26.10, http://broadinstitute.github.io/picard/). Germline genetic variations found in the population were removed using three datasets: HapMap, 1000 genomes phase 3, and National Heart Lung and Exome Sequencing Project data[62]. We removed indels using bcftools view (option "--types snps") (v1.9)[88]. Where variants with multiple alleles existed, one was selected at random. 48,161 putative somatic mutations were identified across 37,482 enhancers in prostate cancer. An average of 1.46 mutations was observed per enhancer (0.08% of total sequence length).

## Reporting summary

Further information on research design is available in the Nature Portfolio Reporting Summary linked to this article.

## Data availability

Processed datasets used in this study are available at Zenodo[89]. Human and mouse-specific ChIP-seq original data are available at ArrayExpress (https://www.ebi.ac.uk/biostudies/arrayexpress) under accessions E-MTAB-2633 and E-MTAB-7127. H9 DNA replication time is available in NCBI's Gene Expression Omnibus through GEO Series accession number GSE137764. Mouse PGC and SSC DNA replication times are available under GEO Series accession number GSE109804. Mouse somatic DNA replication times are available under GEO Series accession number GSE18019. ChIP-seq data from the prostate cancer cell line and prostate epithelial cells (PrEC) used in the study are available under GEO Series accession numbers GSE73783 and GSE57498. ChIP-seq data of histone marks in the breast cancer cell line, MCF-7, and healthy epithelial breast cells are available under GEO Series accession numbers GSE96352, GSE86714, GSE139697, and GSE139733, respectively. Matched ATAC-seq from cancer and healthy thyroid samples from three individuals are available under GEO Series accession numbers GSE162515 (C1, C7, C8). ATAC-seq files for the matched pre-leukemic and blast cells from three individuals are available under GEO Series accession number GSE74912 (SU484, SU501, and SU654).

## Code availability

The code used is available at https://github.com/ewonglab/enhancer_turnover[90].

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

## Acknowledgements

We thank M.Roller and P.Flicek for assistance with metadata access, J.Wong, C.Du, S.Clark, and S.Mahony for helpful discussions, Q.Wang for assistance with data processing, and C.Liang for assistance with figure making. P.C.-P. is supported by a UNSW International Postgraduate Scholarship. V.P. is supported by an Australian Government Research Training Stipend Scholarship. E.S.W. is supported by an NHMRC Investigator Grant (GNT2009309), ARC Discovery Project (DP200100250), and a Snow Fellowship.

## Author contributions

P.C.-P., X.Z., V.P., R.S.Y., and E.S.W. contributed code and analyzed data. P.C.-P. contributed to manuscript preparation. E.S.W. conceived and supervised the work and wrote the manuscript.

## Competing interests

The authors declare no competing interests.
