## [Peer Review File · Nature Communications]

Emergence of enhancers at late DNA replicating regionsREVIEWER COMMENTS

Reviewer #1 (Remarks to the Author):

In this study, Cornejo-Páramo and collaborators aim to explore the impact of genomic organization on enhancer turnover rates. While I believe the work is interesting, the results and overall logic are unclear, making it challenging to follow and not easily accessible to a broad audience.

My primary concern is the lack of clarity regarding why the authors chose this particular layer of genomic features to investigate enhancer turnover rates. The work, as it stands, appears to be a fishing expedition without a clear working hypothesis. This is evident in the presentation of results, where each layer of genomic organization is treated as an independent element without clear connections. Additionally, it remains unclear why the authors chose to investigate DNA replication time, chromatin structure, and DNA substitutions over other factors (e.g., length, distance from the starting site, nucleotide composition, number of enhance per gene- if there are many enhancers per genes one would predict a higher turnover rate).

The introduction is currently brief, requiring more background to make the work accessible. It should explicitly explain the goal and rationale behind selecting specific genomic features.

Several minor points need clarification.

Are the data from cat, dog, horse, macaque, marmoset, opossum, pig, rabbit, and rat also from the germline? From which tissue do the ATAC-seq peaks come? The activity of enhancers is dynamic, so it is crucial to compare the right tissue at the correct developmental stage.

Given that the authors are writing for a general journal, a bit more description is required. For example, what is Repli-Seq? how does it work?

Finally, there appears to be a significant amount of awkward empty space on each page; could you clarify the reason for this?

Reviewer #2 (Remarks to the Author):

In this manuscript, Cornejo-Páramo et al. investigate the relationship between replication timing and enhancer evolution. They first show that recently gained enhancers across species and organs are enriched for late replication in the germ line. These late-replicating, recently gained enhancers show a higher number of mutations that are predicted to lead to increased TF binding compared to their inactive orthologs. Recently gained enhancers are more tissue-specific and show weaker evidence of selection compared to conserved enhancers. They are also enriched for AT-rich motifs recognised by tissue-specific TFs. In the final part of the manuscript, the authors show that replication timing is also associated with enhancer gains and losses during cancer evolution.

The manuscript is well written and explores an interesting mechanism for the emergence of new enhancers, a major source of evolutionary innovation. The methods used are well established and well documented. While most of the findings presented in figures 2-4 are largely expected and recapitulate previous reports for evolutionary young enhancers (e.g., PMID: 33602314 and PMID: 37104612), the authors should be lauded for their efforts to thoroughly characterize these late-replicating regions. The parallels drawn between mammalian and cancer evolution, which as noted by the authors operate in completely different time scales, are also very interesting.

My main concerns are the following:

1. Given that enhancers in this study are solely defined by histone marks, and that late-replicating regions show weaker constraint within species, it is debatable whether these regions are real enhancers (i.e., they lead to increased transcriptional output). Several studies have performed reporter assays for candidate enhancers in liver-related cell types (e.g., PMID: 30045748 or <https://www.biorxiv.org/content/10.1101/2022.12.08.519575v1.full>) that the authors could use to assess whether ChIP-seq peaks in late-replicating regions are as likely to activate transcription as those in early-replicating regions. Alternatively, the authors should discuss more prominently the possibility that many of these newly gained ChIP-seq peaks in late-replicating regions might have no effect on gene expression.
2. Conserved enhancers are identified by the authors based on quite relaxed definitions (2 more other species out of 18, regardless of phylogeny). There are good reasons for this, as the fast turnover can lead to loss of conserved enhancers in some lineages. However, the authors' approach has the caveat of potentially misclassifying human-gained enhancers that independently gained activity in another mammalian lineage as conserved. Recurrent gains of enhancer activity in one organ are more likely to happen in regions that already contain some relevant sequence features – for example they are enhancers in other organs (see PMID: 25411453 for an estimated frequency of enhancer repurposing). These repurposed enhancers would in turn likely be enriched for early replicating regions (given that they already show conserved activity in other organs). Thus, it is important for the authors to show that their results are consistent across different thresholds of defining enhancer conservation and not simply due to a relative depletion of enhancers with repurposed activity versus gains in previously inactive regions (or alternatively discuss the caveats of their approach).
3. The authors alternate between using a sequence-based model and motif scanning to predict TF binding in enhancers. It would be best to be consistent across analyses, and ideally show that the results hold using both metrics. Especially with respect to the analyses presented in Figure 3, would the authors detect more evidence for selection if they were to use the sequence-based model to quantify constraint in putative TF binding sites?
4. Related to the previous point, the statement in P6L12: "Our results suggest candidate enhancers are largely neutrally evolving in the human population regardless of their evolutionary history, even though specific motifs may be under selective constraint" seems misleading given that the authors do in fact detect differences (albeit small) in constraint between conserved and recently evolved enhancers.
5. Given that late-replicating enhancers are evolutionary younger and that younger enhancers tend to be more tissue-specific (PMID: 33602314), it is unclear whether the associations reported in Figure 4 are direct or indirect. Comparing the tissue specificity of early and late replicating regions while conditioning for evolutionary age (e.g., separately for conserved and species-specific enhancers) could provide some clues about whether replication timing is directly linked to tissue-specificity or whether this is a consequence of a young evolutionary age.
6. I am not fully convinced by the authors interpretation of the TF motifs enriched in late replicating regions (Figure 5).
 - First, the visualisation choice for Fig. 5D and S12,13 is misleading, as the authors present scaled nucleotide composition for data that are highly interdependent (when GC is higher AT is by definition lower). Thus, a GC bias of 80-20 would look identical to that of 51-49 in such a plot. The same is true for the motifs in Fig. 5E. It would be best to show the actual frequency of nucleotides in early and late-replicating regions to allow the reader to assess the true magnitude of the effect.
 - Given that homeodomain TFs are overall involved in the regulation of rather conserved developmental programs, it is quite surprising that they would show a striking enrichment for late-replicating, recently gained enhancers. Given the discrepancy between the analysis of motifs and in vivo data (which the authors discuss only very briefly), it is worth asking whether the enrichment is due to actual homeodomain binding sites or random AT-rich short sequences that resemble homeodomain motifs but are not actually bound by TFs. If the authors were to focus on conserved (putatively functional) TF motifs, would they still see an enrichment of homeodomain TFs in late-replicating regions? Furthermore, are homeodomain TF motifs enriched in late-replicating enhancers also when compared to (GC content matched) random genomic background or only when compared to

early-replicating enhancers?

Minor comments

- The pattern shown in Fig. 1H is very interesting, as it would suggest that differences in enhancer conservation across organs arise mainly due to the difference in conservation of late-replicating regions, whereas early replicating regions are equally conserved. Perhaps this can be highlighted/discussed further?
- P1L30: "The DNA replication timing program, defined by the temporal order of DNA replication during the Sphase, is closely linked to the spatial organization of chromatin in the nucleus and transcriptional activity". Perhaps it's best to already mention here that late replicating regions are associated with heterochromatin and tissue-specific expression. This is eventually mentioned in P1L45, but since this is addressed to a general audience it could help the reader to clarify this earlier in the introduction.
- P3L52: "Excluding enhancers overlapping TE (~55% of enhancers) only slightly reduced the slope between enhancer turnover and replication time ($p = 6.0 \times 10^{-3}$)" Perhaps it would be more informative to mention the R2 values with and without TEs rather than the P-value here.
- P5L11: "A sequence model allows us to discount the broader chromatin context by focusing on local variant effects based on sequence differences between orthologous regions." I am not sure I agree with this, as one could argue that the broader chromatin context is often shaped by TF binding caused precisely by these variants. The important point here is that sequence-based models are thought to better reflect TF binding than PWM scanning, perhaps it's best to focus on that.
- P7L15: typo: "we showed [that] enhancers"
- P7L19: why is the data not shown?
- P8L3: The term "GC isochore" is introduced for the first time. Given that this is to be published in a journal targeted to a general audience, it would help to define the term before using it.
- Figure 4: "Tissue-specificity enhancers" should probably read "tissue-specific"
- There are several typos in Figure 6: e.g., unchange -> unchanged and quintile -> quantile

Reviewer #3 (Remarks to the Author):

This is a potentially interesting submission addressing the fundamental link between chromatin accessibility, replication timing and gene expression. The studies reported in the paper identified recently-evolved enhancers in mammalian cells, characterizing both tissue-specific enhancers activated during healthy development and novel enhancers emerging in cancer cells. The paper reports analyses of the distribution of both recently-evolved and conserved enhancers in the context of replication timing domains. The observations are interpreted to suggest a preferred enrichment of recently-evolved enhancers in late-replicating genomic regions.

The paper in its current form utilizes appropriate methodology to process and analyze data obtained primarily from public sources. The cancer evolution data analysis is interesting and could be a good start. The use of chromatin modifications (H3K27ac, H3K4me1, no H3K4me3) as enhancer markers and the stratification of DNA sequences by replication timing is appropriate. The identified enhancers would be of interest as a resource for follow-up studies. Some of the interpretations, however, need further elucidation, and the data sources used for the presented analyses should be clarified. Some examples are listed below.

Comments and suggestions:

-The data are interpreted to imply that recently-evolved enhancers have a higher proportion of late-replicating sequences, but as currently written, it seems that the data compare tissue-specific enhancers in a variety of tissues to the replication time in embryonic and sperm cells. It is unclear if those enhancers were located in late-replicating domains in tissue where they are active. To avoid confusion, the abstract should clarify the tissue of origin in which replication timing was determined.

-Related to the above, the paper would benefit from modifying figure legends to include an accurate description of the sources of combined data. For example, in figure 1E, does the "mean replication time" encompass sequence domains from all three 3 germ cell lines that were analyzed separately in Figure 1C? Does "mean replication time" Figure 1F refer to germline replication timing, or is it based on the entire cohort of 22 cell lines?

-The Methods section lists a single ESC line as the source of human replication timing data. Is this correct?

- Since a large portion of chromatin domains alter their replication time during development and differentiation, the figure legends and Results section should explicitly indicate whether domains that exhibited flexible replication time were included or excluded when mean replication time was calculated. What was the fraction of recently-evolved enhancers in flexible regions compared to constant, early or late, regions?

- Figure 4 presents potentially interesting data about the tissue specificity of enhancers. The number of enhancers was provided for each analysis, and it would be useful to indicate how many tissues were used in the analysis. Does the analysis relate to the four mouse tissues described in the text? For this figure, again, it would be useful to indicate the basis of the assigned replication time, and if the conclusion that "tissue-specific enhancers were more likely to be late replicating" refers to late replication in germline or early embryogenesis.

REVIEWER COMMENTS

Reviewer #1 (Remarks to the Author):

In this study, Cornejo-Páramo and collaborators aim to explore the impact of genomic organization on enhancer turnover rates. While I believe the work is interesting, the results and overall logic are unclear, making it challenging to follow and not easily accessible to a broad audience.

We are grateful for the useful feedback. We have clarified and reworded the Introduction and included additional details in Results, figure legends and Discussion to ensure the work is accessible to a broad audience.

My primary concern is the lack of clarity regarding why the authors chose this particular layer of genomic features to investigate enhancer turnover rates. The work, as it stands, appears to be a fishing expedition without a clear working hypothesis. This is evident in the presentation of results, where each layer of genomic organization is treated as an independent element without clear connections. Additionally, it remains unclear why the authors chose to investigate DNA replication time, chromatin structure, and DNA substitutions over other factors (e.g., length, distance from the starting site, nucleotide composition, number of enhancers per gene- if there are many enhancers per genes one would predict a higher turnover rate).

We focused on DNA replication time as it has been directly linked to mutational rates across the genome, and we are interested in potential causes for the formation of new enhancers, particularly in the context of heritable mutations. While distance and the number of enhancers per gene are interesting research topics (e.g., studied in PMID:30612741 and PMID:33310749), these features are not typically expected to be causally associated with causing the formation of new distal cis-regulatory elements.

The introduction is currently brief, requiring more background to make the work accessible. It should explicitly explain the goal and rationale behind selecting specific genomic features.

Thank you for your comment. We have added more background and context to the research question in the Introduction to make it more accessible to a general audience.

Several minor points need clarification.

Are the data from cat, dog, horse, macaque, marmoset, opossum, pig, rabbit, and rat also from the germline? From which tissue do the ATAC-seq peaks come? The activity of enhancers is dynamic, so it is crucial to compare the right tissue at the correct developmental stage.

The multispecies histone mark data are from the liver, muscle, and brain, and the chromatin accessibility data is also from the same tissues in adults. The information on the chromatin-accessible data was in Supplemental Table S1. We have now added the relevant methods for the ATAC-seq overlaps in the Methods. In the species evolutionary context, we focus on germline mutations that can contribute to forming enhancers in different tissues.

Given that the authors are writing for a general journal, a bit more description is required. For example, what is Repli-Seq? how does it work?

Thank you for your comment. We have provided more information on how Repli-seq works.

Finally, there appears to be a significant amount of awkward empty space on each page; could you clarify the reason for this?

We provided each subsection of the results with its page. Some sections were shorter, which led to the white space. We have fixed this.

Reviewer #2 (Remarks to the Author):

In this manuscript, Cornejo-Páramo et al. investigate the relationship between replication timing and enhancer evolution. They first show that recently gained enhancers across species and organs are enriched for late replication in the germ line. These late-replicating, recently gained enhancers show a higher number of mutations that are predicted to lead to increased TF binding compared to their inactive orthologs. Recently gained enhancers are more tissue-specific and show weaker evidence of selection compared to conserved enhancers. They are also enriched for AT-rich motifs recognised by tissue-specific TFs. In the final part of the manuscript, the authors show that replication timing is also associated with enhancer gains and losses during cancer evolution.

The manuscript is well written and explores an interesting mechanism for the emergence of new enhancers, a major source of evolutionary innovation. The methods used are well established and well documented. While most of the findings presented in figures 2-4 are largely expected and recapitulate previous reports for evolutionary young enhancers (e.g., PMID: 33602314 and PMID: 37104612), the authors should be lauded for their efforts to thoroughly characterize these late-replicating regions. The parallels drawn between mammalian and cancer evolution, which as noted by the authors operate in completely different time scales, are also very interesting.

Thank you for the positive feedback and the highly relevant and valuable suggestions for additional analyses. These results were insightful, and we have added their results to the manuscript.

My main concerns are the following:

1. Given that enhancers in this study are solely defined by histone marks, and that late-replicating regions show weaker constraint within species, it is debatable whether these regions are real enhancers (i.e., they lead to increased transcriptional output). Several studies have performed reporter assays for candidate enhancers in liver-related cell types (e.g., PMID: 30045748 or <https://www.biorxiv.org/content/10.1101/2022.12.08.519575v1.full>) that the authors could use to assess whether ChIP-seq peaks in late-replicating regions are as likely to activate transcription as those in early-replicating regions. Alternatively, the authors should discuss more prominently the possibility that many of these newly gained ChIP-seq peaks in late-replicating regions might have no effect on gene expression.

Thank you for the suggestion to interrogate this key aspect of the research question. We have overlapped the human MPRA data from PMID: 30045748 with our enhancer set (unfortunately, we could not access data from the biorxiv paper). We used those enhancers that overlap tested regions and compared the normalized activity score between recent and conserved human liver enhancers. We did observe slightly lower activity as measured by MPRA at late replication time, although this was not statistically significant ($\alpha=0.05$). Please find the plot below.

Although the result may suggest that late replicating enhancers could be less functional, we are also cautious about this interpretation for two reasons. Firstly, developmental enhancers have been shown to show low-affinity binding (e.g., PMID:25557079, 27155014, 26472909). So late replicating enhancers, which appear enriched for developmental function, could be functional by modulating weak enhancer activity, particularly during development.

Secondly, if MPRA activity is causally related to function and organismal fitness, we may expect to see evidence of selection for MPRA activity. However, we did not find consistent evidence linking MPRA activity to evidence of purifying selection. We tested for purifying selection at MPRA-tested enhancers using the human DECODE data to estimate the proportion of rare vs common variants, as per Figure 3. Here, a positive $\text{log}_2(\text{OR})$ reflects more rare and less common variants, which can be taken as evidence for purifying selection. We compared these to inactive enhancers, those tested for MPRA activity but did not make the activity cut-off imposed by the authors of the MPRA study.

Purifying selection was stronger for conserved than recent enhancers (shown previously in Figure 3). However, purifying selection at conserved enhancers that show MPRA activity was no different from conserved enhancers that do not show MPRA activity. A similar pattern was observed for recently evolved enhancers, where MPRA activity did not distinguish between the two groups of sequences.

We have added a discussion of these points, including the new analyses, to the manuscript, including a note that we cannot rule out that late replicating peaks may have a lesser effect on expression but how MPRA activity relates to functional importance and fitness may be complex as new enhancers in late-replicating regions may have subtle functional roles.

2. Conserved enhancers are identified by the authors based on quite relaxed definitions (2 more other species out of 18, regardless of phylogeny). There are good reasons for this, as the fast turnover can lead to loss of conserved enhancers in some lineages. However, the authors'

approach has the caveat of potentially misclassifying human-gained enhancers that independently gained activity in another mammalian lineage as conserved. Recurrent gains of enhancer activity in one organ are more likely to happen in regions that already contain some relevant sequence features – for example they are enhancers in other organs (see PMID: 25411453 for an estimated frequency of enhancer repurposing). These repurposed enhancers would in turn likely be enriched for early replicating regions (given that they already show conserved activity in other organs). Thus, it is important for the authors to show that their results are consistent across different thresholds of defining enhancer conservation and not simply due to a relative depletion of enhancers with repurposed activity versus gains in previously inactive regions (or alternatively discuss the caveats of their approach).

Thank you for raising this important point. We have repeated the evolutionary data analyses using stricter definitions of conservation in human and mouse and referenced the rationale for doing so. We have repeated analyses in Figure 1 and 3 and found highly consistent results at different thresholds for calling conservation.

Using the human population data, we used stricter definitions of conservation in Figure 3. Consistent with expectation, as the number of species required for defining conservation increased, there was a significant increase in the level of purifying selection detected. This was seen for both enhancers and promoters. As expected, promoters were better conserved than enhancers (please see below).

In comparison, recently evolved enhancers showed significantly lower levels of purifying selection than their conserved counterpart. We have added this to Figure 3.

For mouse data, we have reanalyzed the data defining conservation as the presence of orthologous peaks for 5 or more species (including mouse). We find extremely similar trends to previous findings (in Figure 1), where conservation was originally defined using at least 3 species (including mouse). Below is a comparison of the original heatmap with 3 or more species (top) with the results generated by a stricter definition of conservation of 5 or more species (bottom).

Below are reanalyses of Figure 1 panels using a stricter definition of conservation of 5 or more species. These show highly concordant patterns. We have added the plots using the stricter definition of conservation to Supplementary Figures.

3. The authors alternate between using a sequence-based model and motif scanning to predict TF binding in enhancers. It would be best to be consistent across analyses, and ideally show that the results hold using both metrics. Especially with respect to the analyses presented in Figure 3, would the authors detect more evidence for selection if they were to use the sequence-based model to quantify constraint in putative TF binding sites?

We have rerun the selection analysis based on sequences used in the deep learning model. In this model, we tried to address a specific question: is there evidence that mutations in TF binding sites can casually be associated with turnover? In this setting, we only used recently evolved enhancers that could be aligned between human and mouse. In our selection analysis, we observed a trend that indicated enhancers with CEBPA and HNF4A motifs showed slightly higher levels of purifying selection (although this was not significant based on the 95% CI) than those enhancers where the motifs were not predicted. Both groups of enhancers show a higher DAF ratio than all recent enhancers taken together. These regions are conserved in sequence between human and mouse, so they may have other functions or are in linkage with important sequences, which may explain their higher DAF OR . For example, they may be enhancers in different cell types.

Please see Figure 3D for the full panel. We have also added a discussion of these points to the manuscript.

4. Related to the previous point, the statement in P6L12: “Our results suggest candidate enhancers are largely neutrally evolving in the human population regardless of their evolutionary history, even though specific motifs may be under selective constraint” seems misleading given that the authors do in fact detect differences (albeit small) in constraint between conserved and recently evolved enhancers.

Thank you for the comment. We agree this can be misinterpreted and have reworded this sentence. It now reads, “Our results reveal reduced purifying selection at recent enhancers relative to evolutionarily conserved enhancers and recent promoters.” We have included our multi-level conservation analysis discussed above, which does show increased purifying selection with increased conservation across species in Figure 3.

5. Given that late-replicating enhancers are evolutionary younger and that younger enhancers tend to be more tissue-specific (PMID: 33602314), it is unclear whether the associations reported in Figure 4 are direct or indirect. Comparing the tissue specificity of early and late replicating regions while conditioning for evolutionary age (e.g., separately for conserved and species-specific enhancers) could provide some clues about whether replication timing is directly linked to tissue-specificity or whether this is a consequence of a young evolutionary age.

Thank you for the suggestion. This is an interesting question, and we have performed the suggested analysis to dissect the relationship between evolutionary age and tissue specificity. Please find the plot below. Although recent enhancers tend to show a higher proportion of tissue-specific than conserved enhancers, late replication time is associated with increased tissue specificity regardless of evolutionary age. We have added this plot to Figure 4A.

6. I am not fully convinced by the authors interpretation of the TF motifs enriched in late replicating regions (Figure 5).

- First, the visualisation choice for Fig. 5D and S12,13 is misleading, as the authors present scaled nucleotide composition for data that are highly interdependent (when GC is higher AT is by definition lower). Thus, a GC bias of 80-20 would look identical to that of 51-49 in such a plot. The same is true for the motifs in Fig. 5E. It would be best to show the actual frequency of nucleotides in early and late-replicating regions to allow the reader to assess the true magnitude of the effect.

We now also present the data as bar plots of GC content for Figure 5D, E, S14 and 15.

- Given that homeodomain TFs are overall involved in the regulation of rather conserved developmental programs, it is quite surprising that they would show a striking enrichment for late-replicating, recently gained enhancers. Given the discrepancy between the analysis of motifs and in vivo data (which the authors discuss only very briefly), it is worth asking whether the enrichment is due to actual homeodomain binding sites or random AT-rich short sequences that resemble homeodomain motifs but are not actually bound by TFs. If the authors were to focus on conserved (putatively functional) TF motifs, would they still see an enrichment of homeodomain TFs in late-replicating regions?

We have made bar plots for the homeodomain TFs using only HOMER's top quintile of motifs of each homeodomain PWM. We find restricting to the top-scoring motifs did not change the observed trend (Figure 5E, a larger version with motif IDs is available in Figure S16), suggesting the motif patterns identified as enriched do resemble the consensus sites. The discrepancies between the in vivo data and the sequence motifs results can be due to multiple factors. Only binding sites were interrogated and may not reflect enhancer locations, and the milieu of TFs that are present may also be important, as binding cooperativity and enhancer sequence grammar may also contribute to these differences. We have added these points to the manuscript.

Furthermore, are homeodomain TF motifs enriched in late-replicating enhancers also when compared to (GC content matched) random genomic background or only when compared to early-replicating enhancers?

Thank you for the suggestion. We have rerun the analysis sampling random GC-matched sequences to the candidate enhancers (Figure 5E, S16). The overall trend for homeodomain TF motifs remained largely unchanged, suggesting the enrichment for certain TF binding motifs, being well predicted by replication time, can be largely attributed to neutral evolution. However, replication timing did not explain all enriched motifs (e.g., HOXC13, HNF1B, POU4F3), suggesting some motifs possess a different nucleotide composition than that predicted by replication time. This may be suggestive of natural selection (Fig. S16). We have added these points to the manuscript.

Minor comments

- The pattern shown in Fig. 1H is very interesting, as it would suggest that differences in enhancer conservation across organs arise mainly due to the difference in conservation of late-replicating regions, whereas early replicating regions are equally conserved. Perhaps this can be highlighted/discussed further?

Thank you. We have highlighted the disparity in enhancer turnover rates between organs at late-replicating regions. We also connect this finding to relevant research in the Discussion, specifically lineage-specific expression patterns.

- P1L30: “The DNA replication timing program, defined by the temporal order of DNA replication during the Sphase, is closely linked to the spatial organization of chromatin in the nucleus and transcriptional activity”. Perhaps it’s best to already mention here that late replicating regions are associated with heterochromatin and tissue-specific expression. This is eventually mentioned in P1L45, but since this is addressed to a general audience it could help the reader to clarify this earlier in the introduction.

Thank you for the suggestion, and we have moved this up to improve the structure.

- P3L52: “Excluding enhancers overlapping TE (~55% of enhancers) only slightly reduced the slope between enhancer turnover and replication time ($p = 6.0 \times 10^{-3}$)” Perhaps it would be more informative to mention the R2 values with and without TEs rather than the P-value here.

Thank you for the suggestion, we have changed this.

- P5L11: “A sequence model allows us to discount the broader chromatin context by focusing on local variant effects based on sequence differences between orthologous regions.” I am not sure I agree with this, as one could argue that the broader chromatin context is often shaped by TF binding caused precisely by these variants. The important point here is that sequence-based models are thought to better reflect TF binding than PWM scanning, perhaps it’s best to focus on that.

Thank you for pointing this out; we have reworded it to reflect this.

- P7L15: typo: “we showed [that] enhancers”

Thank you, fixed.

- P7L19: why is the data not shown?

Thank you. We have corrected this and added it to SM as Figure S12.

- P8L3: The term “GC isochore” is introduced for the first time. Given that this is to be published in a journal targeted to a general audience, it would help to define the term before using it.

Thank you, we have clarified this.

- Figure 4: “Tissue-specificity enhancers” should probably read “tissue-specific”

Thank you, fixed.

- There are several typos in Figure 6: e.g., unchange -> unchanged and quintile -> quantile

Thank you, we have fixed ‘unchanged’. We used the term quintile as we divided the distribution into five equal parts.

Reviewer #3 (Remarks to the Author):

This is a potentially interesting submission addressing the fundamental link between chromatin accessibility, replication timing and gene expression. The studies reported in the paper identified recently-evolved enhancers in mammalian cells, characterizing both tissue-specific enhancers activated during healthy development and novel enhancers emerging in cancer cells. The paper reports analyses of the distribution of both recently-evolved and conserved enhancers in the context of replication timing domains. The observations are interpreted to suggest a preferred enrichment of recently-evolved enhancers in late-replicating genomic regions.

The paper in its current form utilizes appropriate methodology to process and analyze data obtained primarily from public sources. The cancer evolution data analysis is interesting and could be a good start. The use of chromatin modifications (H3K27ac, H3K4me1, no H3K4me3) as enhancer markers and the stratification of DNA sequences by replication timing is appropriate. The identified enhancers would be of interest as a resource for follow-up studies.

Thank you for raising this important point, we have now added all datasets to Zenodo [doi:10.5281/zenodo.10494781](https://doi.org/10.5281/zenodo.10494781)

Some of the interpretations, however, need further elucidation, and the data sources used for the presented analyses should be clarified. Some examples are listed below.

Comments and suggestions:

-The data are interpreted to imply that recently-evolved enhancers have a higher proportion of late-replicating sequences, but as currently written, it seems that the data compare tissue-specific enhancers in a variety of tissues to the replication time in embryonic and sperm cells. It is unclear if those enhancers were located in late-replicating domains in tissue where they are active. To avoid confusion, the abstract should clarify the tissue of origin in which replication timing was determined.

Thank you. We have ensured that the tissue of origin of the replication timing data is stated. The tissue of origin of the replication time is the germline cells in mouse and H9 embryonic stem cells in human (as germline data is not available for human). In cancer, replication time for the relevant

cell types were used. We have revised to ensure that this is clear across the manuscript, including in figure legends. This is now also clarified in the Methods.

-Related to the above, the paper would benefit from modifying figure legends to include an accurate description of the sources of combined data. For example, in figure 1E, does the “mean replication time” encompass sequence domains from all three 3 germ cell lines that were analyzed separately in Figure 1C? Does “mean replication time” Figure 1F refer to germline replication timing, or is it based on the entire cohort of 22 cell lines?

Thank you. Mean mouse germline replication time was calculated across PGC (n = 2) and SSP cell lines (n = 2). 1E,F,G,H all show germline replication timing. We have made sure that this is now clear across figure legends.

-The Methods section lists a single ESC line as the source of human replication timing data. Is this correct?

Thank you for the comment. This is correct for our human evolutionary analysis. We used high temporal resolution Repli-seq data in H9 cells from Zhao et al. 2020 (PMID:32209126). This was subjected to mappability corrections to remove noise. The data also was shown to show high concordance with previously produced Repli-seq data but in higher resolution.

Replication time data for the relevant cell lines was used in cancer analyses. While this was included in the main text, we had mistakenly omitted this from Methods. Thank you for pointing this out, and we have now included this information in Methods.

- Since a large portion of chromatin domains alter their replication time during development and differentiation, the figure legends and Results section should explicitly indicate whether domains that exhibited flexible replication time were included or excluded when mean replication time was calculated. What was the fraction of recently-evolved enhancers in flexible regions compared to constant, early or late, regions?

Thank you for the comment. We included all regions where replication time can be determined in downstream analyses. We have now clarified this in the Methods. Across all the cell types in Figure 1C, 21% and 6% of our recently evolved enhancers were defined as consistently early and late replicating, respectively (>0.5 or <-0.5 across all columns of Fig. 1C). We have added a panel to Figure 1 with this information (Fig. 1D).

- Figure 4 presents potentially interesting data about the tissue specificity of enhancers. The number of enhancers was provided for each analysis, and it would be useful to indicate how many tissues were used in the analysis. Does the analysis relate to the four mouse tissues described in the text? For this figure, again, it would be useful to indicate the basis of the assigned replication time, and if the conclusion that “tissue-specific enhancers were more likely to be late replicating” refers to late replication in germline or early embryogenesis.

Thank you. Yes, the analysis relates to the four mouse tissues. We have clarified this in the figure legend and clarified the cell type from which the replication time data is from. In mouse, this was always the average of PGC (n = 2 cell lines) and SSP cell lines (n = 2 cell lines) when we use germline timing. For human, this was H9.

REVIEWERS' COMMENTS

Reviewer #1 (Remarks to the Author):

The authors have convincingly addressed all of my concerns. I believe that this work constitutes an important contribution to the field.

Reviewer #1 (Remarks on code availability):

I haven't rerun the codes, but the GitHub page is accessible, and it seems that the code underlining every figure is well-organized.

Reviewer #2 (Remarks to the Author):

I would like to thank the authors for their rigorous analyses that thoroughly addressed my points. I have no further concerns.

My only very minor point is that I noticed that Fig. S6 appears twice in the supplementary materials.

Reviewer #2 (Remarks on code availability):

The repository is a collection of scripts that document the main analysis steps presented in this manuscript. I did not try running the code, as that would require installation of multiple packages and access to the data, but it seems reasonably well organised and documented.

Reviewer #3 (Remarks to the Author):

The revision has addressed my concerns, no other comments.